# Reptile-like physiology in Early Jurassic stem-mammals

Elis Newham [1,2 ✉], Pamela G. Gill [3,4 ✉], Philippa Brewer [4], Michael J. Benton [3], Vincent Fernandez[5,6], Neil J. Gostling [7], David Haberthür [8,9], Jukka Jernvall[10], Tuomas Kankaanpää [11], Aki Kallonen[12], Charles Navarro[3], Alexandra Pacureanu [6], Kelly Richards[13], Kate Robson Brown [14], Philipp Schneider [2], Heikki Suhonen[12], Paul Tafforeau [6], Katherine A. Williams[2], Berit Zeller-Plumhoff [15] & Ian J. Corfe [10,16 ✉]

Despite considerable advances in knowledge of the anatomy, ecology and evolution of early mammals, far less is known about their physiology. Evidence is contradictory concerning the timing and fossil groups in which mammalian endothermy arose. To determine the state of metabolic evolution in two of the earliest stem-mammals, the Early Jurassic *Morganucodon* and *Kuehneotherium*, we use separate proxies for basal and maximum metabolic rate. Here we report, using synchrotron X-ray tomographic imaging of incremental tooth cementum, that they had maximum lifespans considerably longer than comparably sized living mammals, but similar to those of reptiles, and so they likely had reptilian-level basal metabolic rates. Measurements of femoral nutrient foramina show *Morganucodon* had blood flow rates intermediate between living mammals and reptiles, suggesting maximum metabolic rates increased evolutionarily before basal metabolic rates. Stem mammals lacked the elevated endothermic metabolism of living mammals, highlighting the mosaic nature of mammalian physiological evolution.

[1] School of Physiology, Pharmacology & Neuroscience, University of Bristol, Bristol, UK. [2] Bioengineering Science Research Group, Faculty of Engineering and Physical Sciences, University of Southampton, Southampton, UK. [3] School of Earth Sciences, University of Bristol, Bristol, UK. [4] Earth Sciences Department, The Natural History Museum, London, UK. [5] Core Research Laboratories, The Natural History Museum, London, UK. [6] ESRF, The European Synchrotron, Grenoble, France. [7] School of Biological Sciences, University of Southampton, Southampton, UK. [8] Swiss Light Source, Paul Scherrer Institut, Villigen, Switzerland. [9] Institute of Anatomy, University of Bern, Bern, Switzerland. [10] Institute of Biotechnology, University of Helsinki, Helsinki, Finland. [11] Department of Agricultural Sciences, University of Helsinki, Helsinki, Finland. [12] Department of Physics, University of Helsinki, Helsinki, Finland. [13] Oxford University Museum of Natural History, Oxford, UK. [14] Department of Anthropology and Archaeology, University of Bristol, Bristol, UK. [15] Institute for Materials Research, Division of Metallic Biomaterials, Helmholtz Zentrum Geesthacht, Geesthacht, Germany. [16] Geomaterials and Applied Mineralogy group, Geological Survey of Finland, Espoo, Finland. ✉email: en12630@bristol.ac.uk; pam.gill@bristol.ac.uk; ian.corfe@helsinki.fi

Recent discoveries and analyses have revolutionized our knowledge of Mesozoic mammals, revealing novel aspects of their ecology[1,2], development[2,3], systematics[2,3] and macroevolution[4,5]. However, details of physiology are more more difficult to determine from fossils, and our knowledge of physiological evolution remains comparatively poor. Living mammals are endotherms, possessing the ability to control and maintain metabolically produced heat and have a substantially higher capacity for sustained aerobic activity than ectothermic animals[6–8]. The origin of endothermy is an important event in mammalian evolution, often noted as key to their success[6–8]. There are a number of competing evolutionary hypotheses for the origin of endothermy: (a) selection for higher maximum metabolic rates (MMRs) enhanced sustained aerobic activity[6,9,10], (b) selection for higher basal metabolic rates (BMRs) enhanced thermoregulatory control[11,12], or (c) MMRs and BMRs evolved in lockstep with each other[7,8].

Direct evidence from living mammals to support these hypotheses is equivocal[7]. Recent analyses find no long-term evolutionary trend in BMR[13] contradicting earlier suggestions of increasing BMR throughout the Cenozoic[12] and so implying that the Middle Jurassic (~170 Ma) most recent common ancestor (MRCA) of living mammals[13] possessed a BMR within the range of present-day mammals. Several indirect indicators of metabolic physiology in fossil synapsids have been suggested but provide contradictory evidence for the timing of the origin of endothermy and its evolutionary tempo. These include: the presence of fibrolamellar long-bone histology[14,15], first seen in the Early Permian synapsid *Ophiacodon* ~300 million years (Ma) ago[16]; the presence of an infraorbital canal and lack of parietal foramen, used to infer facial whiskers, fur, lactation and endothermy in Early Triassic (~245 Ma) cynodonts[17]; inferred maxillary nasal turbinates in the Late Permian (~255 Ma) therapsid *Glanosuchus*, used to suggest that mammalian levels of endothermy evolved by the Late Triassic (~210 Ma)[18]; a trend towards increased relative brain size initiated in Late Triassic non-mammaliaform cynodonts[19] and the mammaliaform (stem mammal *sensu* Rowe[20]) *Morganucodon*[21,22]; and acquisition of a parasagittal gait in the Early Cretaceous (~125 Ma) therian mammals *Eomaia* and *Sinodelphys*[23]. Several recent studies provide more quantitative links to physiological parameters. Oxygen isotopes were used to infer elevated thermometabolism in Middle–Late Permian (~270–255 Ma) eucynodonts[24], red blood cell size diminution in Late Permian (~255 Ma) eutheriodontid therapsids was linked via two proxies to increased MMR[25] and osteocyte lacuna shape correlations suggested "mammalian" resting metabolic rates in Permo-Triassic (~250 Ma) dicynodonts[26].

However, the inconsistency of these characters, in time and with respect to phylogeny[27,28], along with re-assessments of function in relation to endothermy[7,29,30], limit their use as conclusive indicators of modern mammalian levels of endothermy in fossil taxa. Such temporal and phylogenetic heterogeneity suggests that the evolution of mammalian endothermy followed a complex, mosaic pattern with different physiological aspects likely evolving independently, and at separate rates, towards current mammalian levels. Additionally, few of these physiological proxies are directly related to measurable aspects of metabolic rate.

To address these issues, we use two proxies to improve understanding of physiology at one of the most important nodes along this transition. We do so by estimating BMR and growth rate, and calculating a known proxy for MMR, for two of the earliest known mammaliaforms, *Morganucodon* and *Kuehneotherium*[1,31]. Using cementochronology to estimate maximum lifespan by counting growth increments in synchrotron radiation-based micro-computed tomographic (μCT) data of fossil dental cementum, we estimate that both taxa had significantly longer lifespans than extant mammals of comparable size. By regressing lifespan against BMR and postnatal growth rate in extant mammals and reptiles, we in turn estimate significantly lower values for both of these metrics for the earliest mammaliaforms. However, when we compare the blood flow index (the ratio between femoral nutrient foramina area and femur length that serves as a proxy for MMR) of *Morganucodon* with those of extant taxa, we find that *Morganucodon* had an intermediate value between living mammals and reptiles. These results suggest that basal mammaliaforms occupied a metabolic grade similar to living reptiles and had yet to achieve the endothermic physiology of living mammals.

## Results

**Lifespan as a proxy for mammaliaform physiology.** We used maximum lifespan (i.e. the single longest known lifespan of a taxon) estimates for fossil mammaliaform taxa as a proxy for both BMR[32] and postnatal growth rate[33]. In extant tetrapods, negative correlations exist between maximum lifespan and BMR[32] and between maximum lifespan and growth rate[33,34]. In general, the longer a mammal's lifespan, the lower its size-adjusted BMR and growth rate. Growth rates have been shown to correlate strongly with metabolic power in extant vertebrates, with endotherms growing an order of magnitude faster than ectotherms[34,35]. Maximum lifespan is an applicable value for fossil samples, as, unlike other metrics (e.g. 10% most long lived or mean lifespan of a cohort), it does not rely on cohort- or population-based statistics that fossil samples cannot fulfil[36]. This value is also less susceptible to extrinsic population-level factors on lifespan, such as disease or predation, and relates most closely to the physiological limit of lifespan of an organism. An accurate assessment of maximum lifespan in fossil mammals can therefore be used to estimate their metabolic potential.

To estimate mammaliaform lifespans, we used cementochronology. This well-established technique, which counts annual growth increments in tooth-root cementum, has been used to record lifespans in extant mammals[37] with >70 species aged using this technique[38]. Cementum is a mineralized dental tissue surrounding the tooth root (Fig. 1a, b), attaching it to the periodontal ligament and anchoring the tooth within the alveolus[37]. Growth of cementum is continuous throughout life in extant mammals and seasonally appositional in nature, forming a series of increments of differing thickness and opacity when viewed in histological thin sections under light microscopy. The correlation between increment count and chronological age is well documented, with one thick and one thin increment deposited every year[38]. It has been shown that the thin, hyper-mineralized opaque increments record growth rate reduction in less favourable seasons[39,40].

Despite this potential, cementochronology has not previously been attempted for fossil mammals older than the Pleistocene (2.6 Ma)[41], because histological thin sections destroy fossils and provide only a restricted field of view. We overcame these problems by using propagation phase-contrast X-ray synchrotron radiation microtomography (PPC-SRμCT) to non-destructively image fossilized cementum increments[42,43]. The sub-micrometre resolution, fast-throughput and three-dimensional (3D) nature of PPC-SRμCT allows for large sample sizes and for increments to be imaged along their entire transverse and longitudinal trajectories in volumetric PPC-SRμCT data sets. Cementum increments are known to occasionally split and coalesce, creating errors in counts based on single, or limited numbers of, two-dimensional thin sections created for each tooth[37,44] (Supplementary Fig. 1). PPC-SRμCT imaging and 3D segmentation of individual increments across extensive vertical distances within

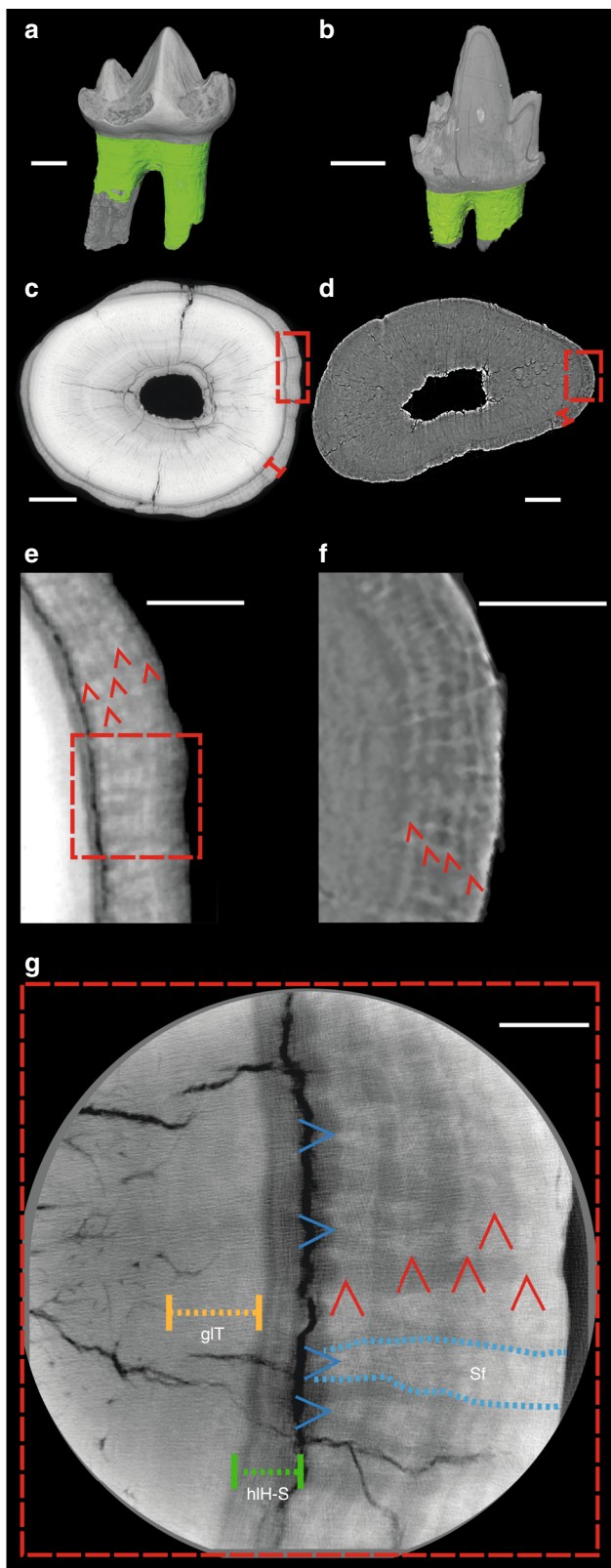

Fig. 1 Cementum of *Morganucodon* and *Kuehneotherium*. **a**, **b** Three-dimensional reconstructions of **a** *Morganucodon* right lower molar tooth NHMUK PV M 104134 (voxel size 2 μm, μCT) and **b** *Kuehneotherium* right lower molar tooth NHMUK PV M 21095 (voxel size 1.2 μm, PPC-SRμCT). Green = cementum. **c**, **d** Transverse PPC-SRμCT virtual thin sections (0.33 μm voxel size) of roots of **c** NHMUK PV M 104134 and **d** NHMUK M 27436. Red bracketed line highlights extent of cementum surrounding dentine. **e**, **f** Close-ups of boxes in **c**, **d**, with five and four circumferential light/dark increment pairs highlighted by red arrows, respectively. **g** Synchrotron nanotomographic virtual thin section of NHMUK PV M 104134 (30 nm voxel size) provides close-up of region close to box in **e**. Vertical red arrows = cementum increments; horizontal blue arrows, dashed blue lines and Sf = radial bands of Sharpey's fibres; yellow dashed line and glT = granular layer of Tomes; green dashed line and hlH-S = hyaline layer of Hopewell-Smith. Scale bars represent 500 μm in **a**, **b**, 100 μm in **c**, **d**, 30 μm in **e**, **f** and 10 μm in **g**.

(Supplementary Notes 1 and 2 and Supplementary Data 1 and 2). Thousands of their bones and teeth were washed into karst fissures that have subsequently been revealed by quarrying. This provides a rare opportunity to analyse large samples of fossil material needed for confident estimation of maximum lifespan. Importantly, these are the earliest diphyodont taxa (Fig. 2), with a single replacement of non-molar teeth and no molar tooth replacement[31], and so estimates of lifespan are accurate to the time of the measured tooth-root formation.

The fossil sample studied included both isolated teeth and mandibles with multiple teeth or roots in situ. We applied PPC-SRμCT to 87 *Morganucodon* specimens (52 isolated teeth, 35 dentaries, all *Morganucodon watsoni*) and 119 *Kuehneotherium* specimens (116 isolated teeth, 3 dentaries) (see "Methods"). From these, 34 *Morganucodon* and 27 *Kuehneotherium* specimens were sufficiently well preserved for three observers to independently estimate lifespan from cementum increments. These estimates were compared to validate their accuracy and precision (see "Methods"; Supplementary Data 1). The remainder showed physical and/or diagenetic damage that prevented increment measurement (Supplementary Fig. 2).

The cementum of *Morganucodon* and *Kuehneotherium* (Fig. 1a, b) is distinguished from dentine in our PPC-SRμCT data by a distinct boundary layer separating the two tissues. This lies external to the granular layer of Tomes of the dentine and is interpreted as the hyaline layer of Hopewell–Smith (Fig. 1c–g). Synchrotron nanotomographic imaging (30 nm isotropic voxel size) highlights individual Sharpey's fibre bundles (linking cementum to the periodontal ligament in extant mammals) visible in several exceptionally preserved specimens, which can be traced radially through the cementum (Fig. 1g). Across the tooth-root transverse axis, cementum is ~10–70 μm radial thickness and displays a series of contrasting light and dark circumferential increments representing different material densities (Figs. 1e–g and 3a–d). Higher-density increments (represented by greater greyscale values) are on average 2–3 μm radial thickness, and lower density increments are 1–3 μm radial thickness (Fig. 1c–g and 3a–d). Individual increments can be followed continuously both longitudinally and transversely through the entire scanned volume of a tooth root (Fig. 3e, f).

**Increment count accuracy, tooth eruption sequence and timing.** We tested the accuracy of cementum increment counts for predicting lifespan in fossils by additional PPC-SRμCT imaging and counting of increments in the cementum of several teeth along the tooth row in eight dentulous *Morganucodon* specimens with a range of teeth in situ (Table 1) and of growth increments (lines of arrested growth (LAGs)) in the periosteal region of the

the cementum allowed us to confidently distinguish principal annual increments from any accessory increments created by lensing and coalescence (see "Methods").

*Morganucodon* and *Kuehneotherium* are shrew-sized insectivores[1], which co-existed on a small landmass during the Early Jurassic marine transgression (Hettangian-early Sinemurian, ~200 Ma) in what is now Glamorgan, South Wales, UK[45]

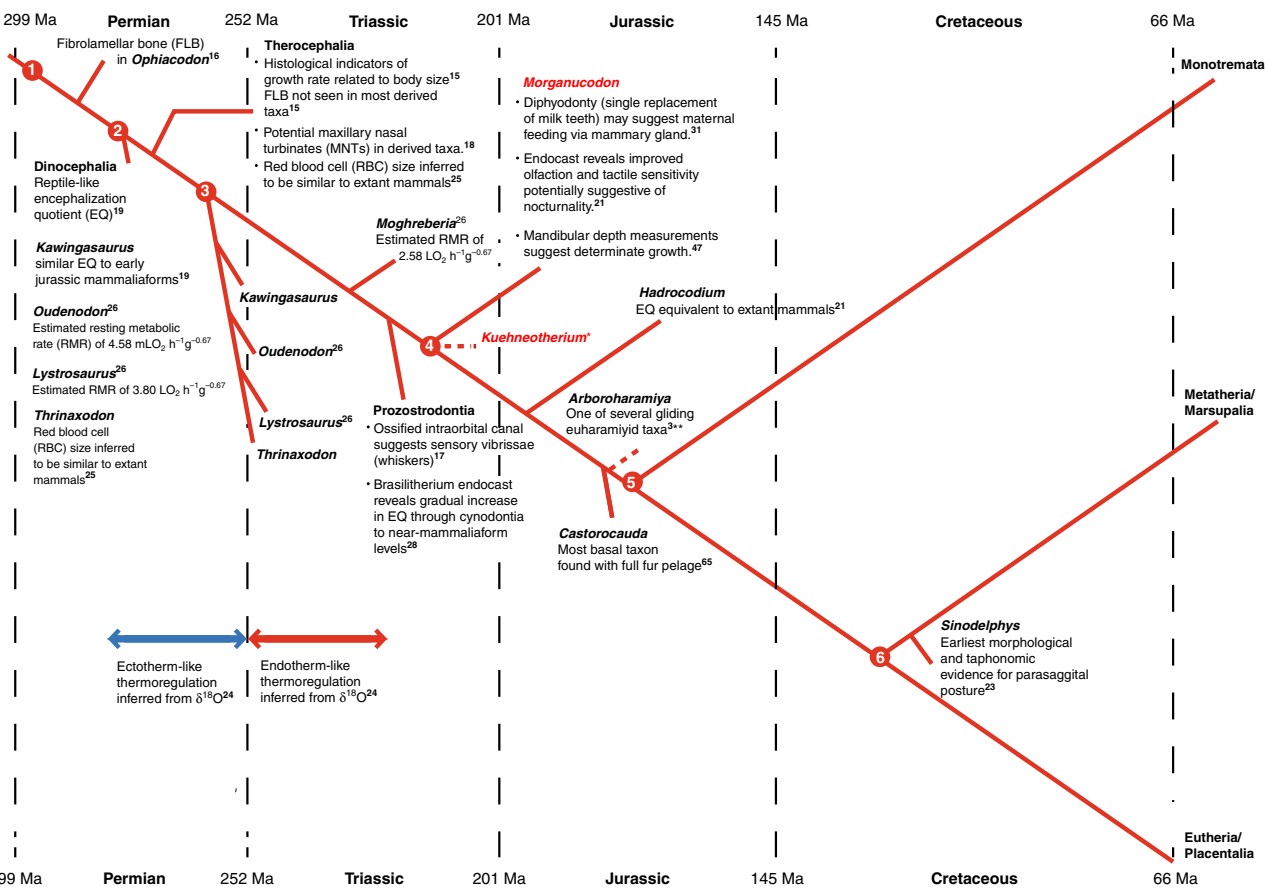

**Fig. 2 Time-scaled phylogeny summarizing evidence for physiological evolution along the synapsid lineage towards mammals.** Red nodes highlight the divergence of major lineages; Node 1 = divergence of the Pelycosaur lineage; Node 2 = the Therapsida clade; Node 3 = the Cynodontia clade; Node 4 = the Mammaliaformes clade; Node 5 = the Mammalia clade; Node 6 = the Theria clade. Superscript numbers denote references in the main text. Single asterisk (*) denotes the uncertain phylogenetic affinities of *Kuehneotherium* within the Mammaliaformes clade[31]. Double asterisks (**) denote the uncertain phylogenetic affinities of *Arboroharamiya*[3].

dentary bone in two of these (Fig. 4). In both specimens where dentary LAGs are found, counts are identical with the cementum increments in the teeth (p3–m2; Fig. 4). Also, comparisons between counts of cementum increments are identical across all four premolars (p1–p4) and the anterior molars (m1–m2), in all specimens where they occur together (Table 1). This agreement between p1–m2 teeth and dentary increment counts indicates that growth in both teeth and jaws was following the same, circum-annual rhythm, as previously reported for multiple extant mammal species[37]. We consider this to be strong support for the accuracy of lifespan estimates based on these increment counts.

The increment counts along *Morganucodon* dentary toothrows can also provide information on eruption sequence and timing (see Supplementary Note 3 for more details). The first permanent premolar to the second molar all erupted within 1 year, with the first molar erupting prior to the third and fourth premolars (NHMUK PV M 27312; see Supplementary Note 3 for more details). The ultimate incisor (i4), the canine and the third molar erupted in the following year. We do not have information on eruption timing of more anterior incisors, or the fourth molar, and the fifth molar is only occasionally present. As we estimate that *Morganucodon* was long lived relative to comparatively sized extant mammals (see below), this pattern of most of the adult tooth row being in place during the first 2 years of life is also supportive of a relatively short (compared with its total lifespan) juvenile stage[46] and determinate growth[47]. The absolute length of these stages in *Morganucodon* is, however, considerably longer

than extant mammals of comparable body size[37]. Unfortunately, dentulous specimens of *Kuehneotherium* are rare, and there are no tooth rows with cementum increment counts in our sample.

**Long lifespans, low BMR and growth rates.** Cementum increment counts provide a minimum estimate of maximum lifespan of 14 years for *Morganucodon* and 9 years for *Kuehneotherium* (Figs. 3 and 5a and Supplementary Data File 1). These may underestimate true maximum lifespan, as any damage to outer cementum increments would reduce estimated maximum lifespan. One-way analysis of variance (ANOVA) comparisons of mean intra-observer coefficient of variation (CV) between our study and ten previous cementochronological studies of different extant mammal species (see "Methods") with similar age ranges suggest that values for PPC-SRμCT data (Shapiro–Wilk $W = 1$) of *Morganucodon* (CV = 9.32) and *Kuehneotherium* (CV = 4.89) are significantly lower than previous thin section-based studies ($W = 0.93$; minimum extant CV = 14.2, mean CV = 21.8, standard deviation = 5.87; $F = 11.12$, $p < 0.01$; Supplementary Table 1).

We estimated body mass ranges of 10.7–25.0 g (mean 17.9 g) for *Morganucodon* and 14.9–32.7 g (mean 23.8 g) for *Kuehneotherium* (minimum mass estimates based on skull length[21] and maximum mass estimates on dentary length[48]; see "Methods"). Maximum lifespan and mean body mass for the mammaliaforms were compared with published data for large

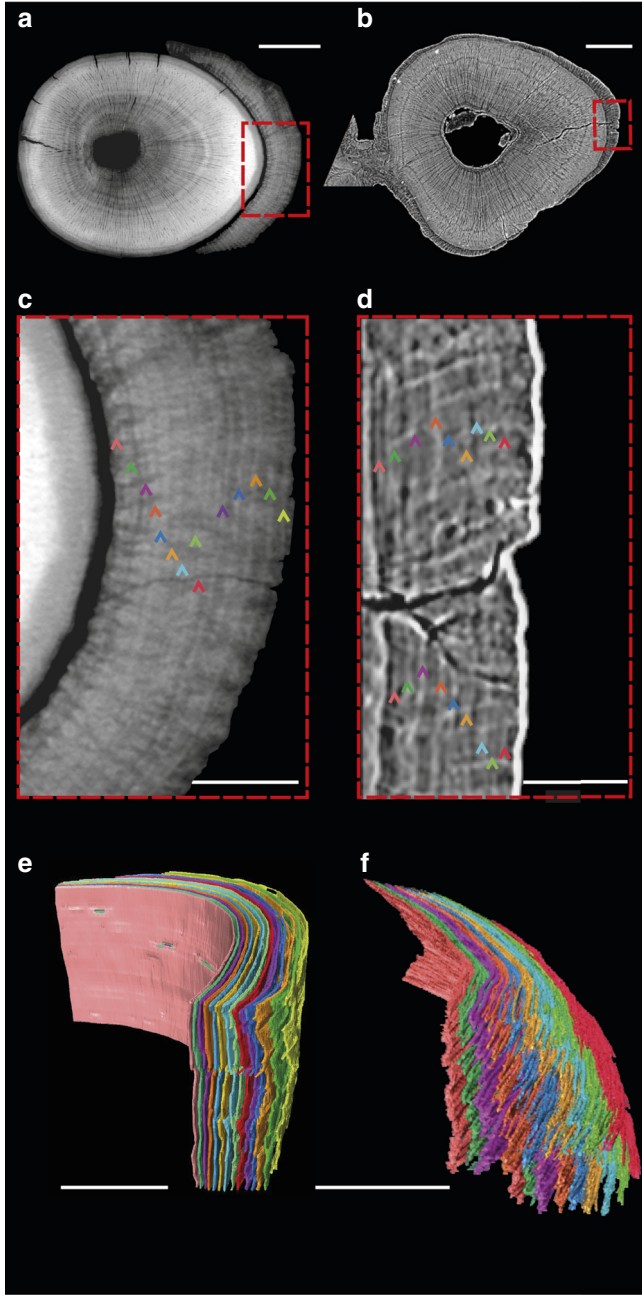

**Fig. 3 Three-dimensional segmentation of *Morganucodon* and *Kuehneotherium* specimens with the highest counts of cementum increments. a, b** Transverse virtual thin sections of PPC-SRµCT reconstructions (0.33 µm³ voxel size). **a** *Morganucodon* specimen NHMUK PV M 104127 showing a 55-µm-thick layer of cementum around the root dentine. **b** *Kuehneotherium* specimen UMZC Sy 141 showing a 32-µm-thick layer of cementum. **c, d** Detail of the cementum of **c** NHMUK PV M 104127 and **d** UMZC Sy 141. Cementum increments highlighted by 14 and 9 multi-coloured arrows, respectively. **e, f** 3D segmentations of the cementum increments of **e** NHMUK PV M 104127 and **f** UMZC Sy 141. The colour of each increment corresponds to the colours of each arrow in **c, d**, respectively. Scale bars represent 100 µm in **a, b**, 30 µm in **c, d** and 30 µm in **e, f**.

**Table 1 Cementum and dentary increment counts for each element of dentulous *Morganucodon* specimens.**

| Specimen | Element | Increments |
|---|---|---|
| NHMUK PV M 95790 | i4 | 7 |
| NHMUK PV M 95790 | c | 7 |
| NHMUK PV M 95790 | p1 | 8 |
| NHMUK PV M 95790 | p2 | 8 |
| NHMUK PV M 95790 | p3 | 8 |
| NHMUK PV M 96413 | p3 | 5 |
| NHMUK PV M 96413 | p4 | 5 |
| NHMUK PV M 96413 | m1 | 5 |
| NHMUK PV M 96413 | Dentary | 5 |
| NHMUK PV M 96396 | p4 | 4 |
| NHMUK PV M 96396 | m1 | 4 |
| NHMUK PV M 96396 | m2 | 4 |
| NHMUK PV M 96396 | m3 | 3 |
| NHMUK PV M 96396 | Dentary | 4 |
| NHMUK PV M 95809 | m1 | 3 |
| NHMUK PV M 95809 | m2 | 3 |
| NHMUK PV M 104128 | m1 | 5 |
| NHMUK PV M 104128 | m2 | 5 |
| NHMUK PV M 96441 | m1 | 5 |
| NHMUK PV M 96441 | m2 | 5 |
| NHMUK PV M 104130 | m1 | 5 |
| NHMUK PV M 104130 | m2 | 5 |
| NHMUK PV M 104130 | m3 | 4 |
| NHMUK PV M 104129 | m1 | 9 |
| NHMUK PV M 104129 | m2 | 9 |
| NHMUK PV M 104129 | m3 | 8 |

The lower dental formula of *Morganucodon* is 4:1:4:4–5[1].

values. To ensure robustness of our results, we additionally analysed maximum captive lifespans of extant taxa below 100 g, which show an average increase above maximum wild lifespans of approximately 3.43 and 4.38 years per taxon for mammals and reptiles, respectively (Supplementary Fig. 4). Broad results of statistical tests, and the overall conclusions of our study, are unchanged regardless of whether wild or captive data are used for analysis and comparisons between our fossil lifespans and the lifespans of extant taxa (see Supplementary Note 4 and Supplementary Figs. 1–6).

Phylogenetic generalized least squares (PGLS) regression of $\log_{10}$-transformed values shows that the fossil mammaliaforms fall within the range of extant reptiles and have longer maximum lifespans for their size and are further above the mammal regression mean, than all extant mammals under 4 kg (the long-lived and secondarily dwarfed[49] mouse lemur *Microcebus murinus* is closest). Only the short-beaked echidna *Tachyglossus aculeatus*, a monotreme with long lifespan and low metabolic rate, exceeds the distance above the mammalian mean for *Kuehneotherium*, but not for *Morganucodon* (Fig. 5b). One-way phylogenetic analysis of covariance (ANCOVA) comparisons show that regression slopes for extant mammals and reptiles are statistically similar ($p = 0.35$) but their means are significantly separated ($p = 0.036$), with reptiles on average living 18.3 years longer than mammals of the same body mass.

To estimate BMR, we used PGLS and recovered significant correlations between $\log_{10}$-transformed values of maximum wild lifespan and mass-specific standard metabolic rate (msSMR; measured in mL $O_2$ h$^{-1}$ g$^{-1}$ and analogous with BMR in extant mammals—SMR was used as BMR cannot be measured in reptiles[50]) from published data for 117 extant mammals and 55 extant reptiles ("Methods"; Supplementary Data 3 and Fig. 6a). Using the correlation between maximum wild reptile lifespan and msSMR and plotting our mammaliaform directly onto this

samples of terrestrial, non-volant wild extant mammal ($n = 278$) and non-avian reptile ($n = 256$) species ("Methods"; Supplementary Data 3). Maximum wild lifespans of extant taxa were chosen for comparison with our fossil taxa, as these values are the closest analogue to our estimated lifespans, relative to captive lifespan

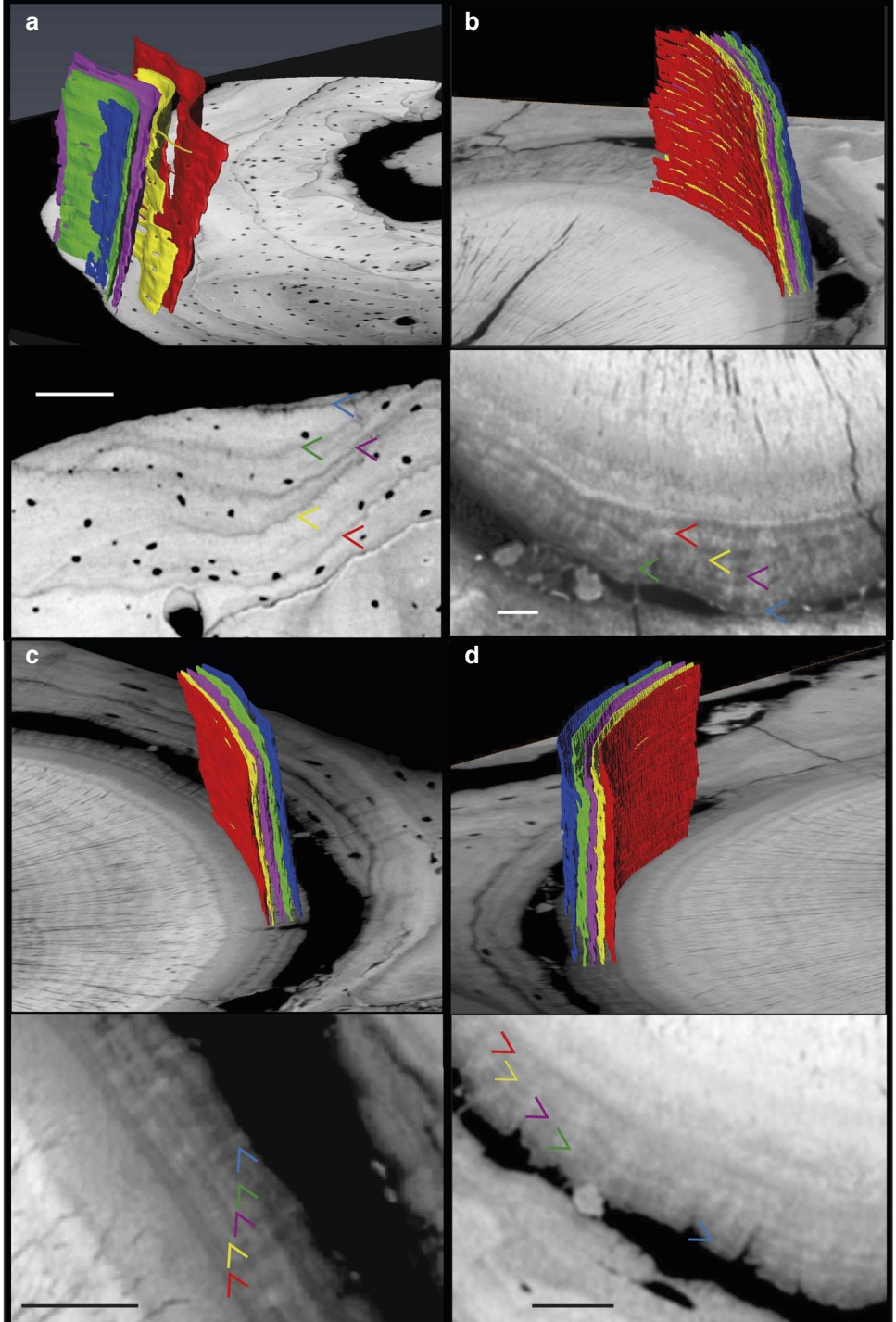

**Fig. 4 Shared increment patterns between m1 and m2 tooth-root cementum and the dentary of *Morganucodon* specimen NHMUK PV M 96413. a** Four lines of arrested growth and a fifth incipient one are visible within the periosteal region of the dentary, each highlighted by three-dimensional segmented bands of differing colour corresponding to coloured arrows in the accompanying transverse PPC-SRμCT slice. Only LAGs persisting through the volume are segmented and highlighted. This pattern is mirrored in **b** the anterior root of the m1 tooth, **c** the posterior root of the same m1 tooth and **d** the anterior root of the m2 tooth. Scale bars represent 30 μm.

regression line, we estimated a reptile-derived msSMR of 0.055 mL $O_2$ $h^{-1}$ $g^{-1}$ (*Morganucodon*) and 0.08 mL $O_2$ $h^{-1}$ $g^{-1}$ (*Kuehneotherium*) (Fig. 6a). We additionally used the correlation between maximum wild mammal lifespan and msSMR and estimated a mammal-derived msSMR of 0.36 mL $O_2$ $h^{-1}$ $g^{-1}$ for *Morganucodon* and 0.46 mL $O_2$ $h^{-1}$ $g^{-1}$ for *Kuehneotherium* (Fig. 6a). When $log_{10}$ PGLS is used to regress these estimates

against body mass, both mammaliaforms fall outside the 95% predictor interval (PI) of the mammalian data and within the reptile range of msSMR, regardless of whether mammaliaform msSMR is estimated from reptilian or mammalian data (Fig. 5c). This suggests that the mammaliaforms had significantly lower msSMR values when compared to extant mammals of similar size. The comparably sized mammal (<100 g) of lowest msSMR is

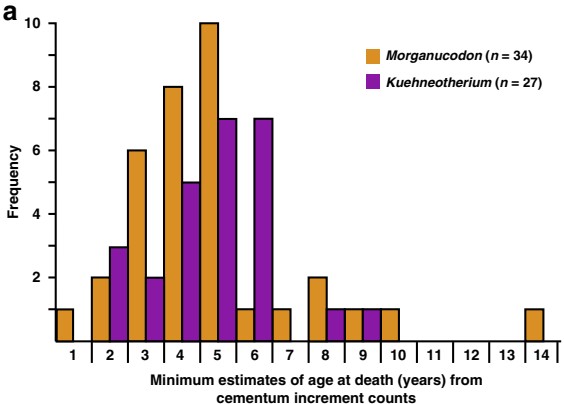

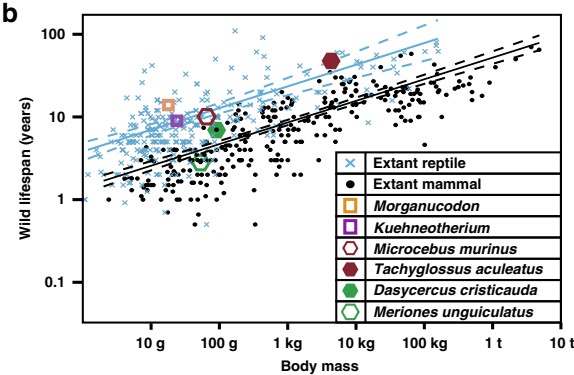

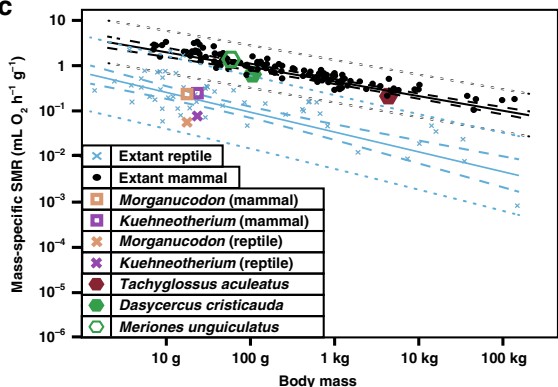

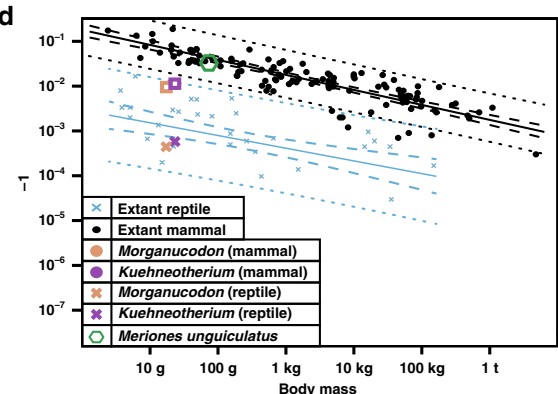

**Fig. 5 Lifespan and metabolic estimates of *Morganucodon* and *Kuehneotherium*. a** Histogram of lifespan estimates from cementum increment counts. **b** $Log_{10}$ phylogenetic least squares (PGLS) biplot of mean body mass (g) against maximum wild lifespan (years) for extant mammals ($n = 279$), extant non-avian reptiles ($n = 252$) and fossil mammaliaforms. **c** $Log_{10}$ PGLS biplot of mean body mass (g) against mass-specific standard metabolic rate (msSMR; mL $O_2$ $h^{-1}$ $g^{-1}$) for extant mammals ($n = 117$) and extant reptiles ($n = 55$) and estimates for fossil mammaliaforms. **d** $Log_{10}$ PGLS biplot of mean body mass (g) against postnatal growth rate constant $K$ (days$^{-1}$) for extant mammals ($n = 115$) and extant reptiles ($n = 33$) and estimates for fossil mammaliaforms. PGLS regression lines in **b**–**d** are shown for extant mammals (black) and extant reptiles (blue), 95% confidence intervals are represented by dashed lines and 95% predictor intervals by dotted lines. Source data are provided as a Source data file.

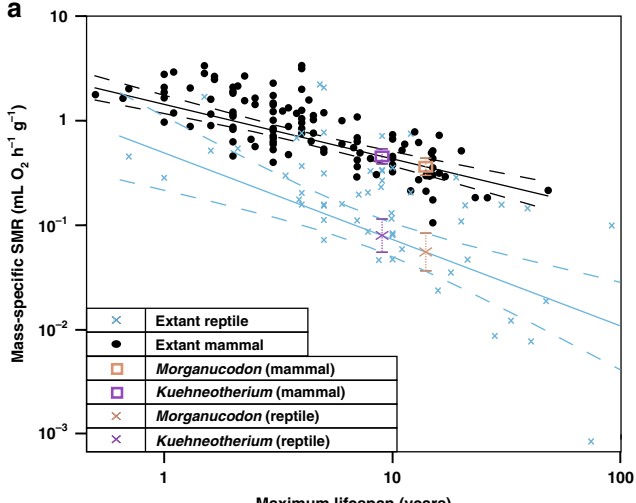

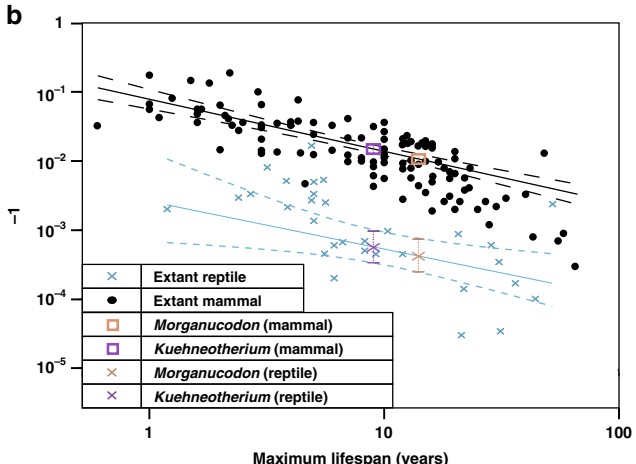

**Fig. 6 The relationship between lifespan and msSMR, and between lifespan and growth rate constant *K*, in mammals and reptiles. a** $Log_{10}$ phylogenetic least squares (PGLS) biplot between maximum wild lifespan (years) and mass-specific standard metabolic rate (msSMR; mL $O_2$ $h^{-1}$ $kg^1$) for extant mammals ($n = 117$) and reptiles ($n = 55$). **b** $Log_{10}$ PGLS biplot between maximum wild lifespan (years) and postnatal growth rate constant $K$ (days$^{-1}$) for extant mammals ($n = 115$) and reptiles ($n = 31$). PGLS regression means for each clade (black lines for mammals, blue lines for reptiles, dashed lines denote 95% confidence intervals) are used to estimate msSMR and $K$ for mammaliaforms *Morganucodon* and *Kuehneotherium*, with dashed brackets denoting their 95% confidence intervals. Source data are provided as a Source data file.

the marsupial *Dasycercus cristicauda*, with a maximum wild lifespan of 7 years and msSMR of 0.63 mL $O_2$ $h^{-1}$ $g^{-1}$ (Fig. 5c).

We estimated growth rates using PGLS correlations between maximum wild lifespan and growth rate[33,34] from published data from 115 extant mammals and 30 extant reptiles ("Methods"; Supplementary Data 3 and Fig. 6b). From mammal data, we

estimate growth rate constants $K$ (days$^{-1}$—see "Methods") of $1.085^{-2}$ days$^{-1}$ (*Morganucodon*) and $1.474e^{-2}$ days$^{-1}$ (*Kuehneotherium*). From reptile data, we estimated $K = 4.91^{-4}$ days$^{-1}$ (*Morganucodon*) and $K = 6.65^{-4}$ days$^{-1}$ (*Kuehneotherium*) (Fig. 6b). Log$_{10}$ PGLS regression against body mass again places both mammaliaforms outside the mammalian 95% PI and within the reptile growth rate range, whether estimated from mammalian or reptilian data (Fig. 5d). The lowest growth rate of any <100 g extant mammal is $K = 3.24e^{-2}$ days$^{-1}$ for the Mongolian gerbil *Meriones unguiculatus*.

In summary, our estimates of maximum lifespan provided by tomographic imaging of cementum increments in *Morganucodon* and *Kuehneotherium* are significantly longer than the maximum wild lifespan of any extant mammal of comparable body mass. These lifespans provide estimates of SMR/BMR and growth rate that are significantly lower than comparably sized extant mammals and instead correspond to those of extant reptiles.

**Femoral blood flow shows intermediate MMR.** To compare our fossil mammaliaform BMR estimates with MMR, we used a second proxy directly linked to MMR[51]. The ratio between nutrient foramen area and femur length has been used as an index for relative blood flow ($Q_i$) through the femur during and after metabolically demanding exercise ($Q_i = r_f^4/L$, where $r_f =$ foramen radius and $L =$ femur length), previously shown to correlate well with MMR[51]. From µCT data of the six most complete *Morganucodon* femoral diaphyses available, we segmented all nutrient foramina (Fig. 7a) and estimated their area by measuring their minimal radii (see "Methods"). *Kuehneotherium* could not be included as no suitable femoral specimens are known.

We estimated a $Q_i$ of $3.829e^{-7}$ mm$^3$ for *Morganucodon* and compared this with published and new data ("Methods") for extant mammals ($n = 69$) and reptiles ($n = 30$). The latter includes varanids ($n = 8$), which in the absence of mammalian predators fill an active hunting niche and tend to have mammalian MMR levels while retaining reptilian BMR levels[51] (Supplementary Table 2). One-way ANCOVA comparisons show that means of GLS regression slopes for extant mammals and non-varanid reptiles are significantly different ($p < 0.01$) while the slopes are similar ($p = 0.16$). Log$_{10}$ GLS regression of body mass and $Q_i$ shows that *Morganucodon* is further above (higher $Q_i$ for its mass) the non-varanid reptile mean than all non-varanid reptiles (phylogenetically informed statistical comparisons were not used here due to the non-significant lambda values showing no phylogenetic signal in the taxa used, see "Methods"). However, *Morganucodon* is also slightly further from the mammalian mean than the non-varanid reptile mean and considerably closer to small non-varanid reptile species data points than those of small mammalian species (Fig. 7b). This intermediate $Q_i$, and so inferred intermediate MMR, suggests that, while retaining typical reptilian BMR and growth rates, *Morganucodon* had MMR above non-varanid reptiles but not as high as mammals or actively foraging varanid reptiles.

## Discussion

We have used two quantitative proxies to determine the metabolic status of early mammaliaforms. Relatively long lifespans for both *Morganucodon* and *Kuehneotherium* result in SMR/BMR and growth rate estimates equivalent to modern reptiles of comparable size and indeed at the higher lifespan/lower BMR/ slower growth end of the reptile scale for *Morganucodon*. This is true whether we compare our fossil estimates to wild lifespans of extant taxa or estimate fossil "captive" lifespans and compare them to captive values for extant taxa. In contrast, femoral blood

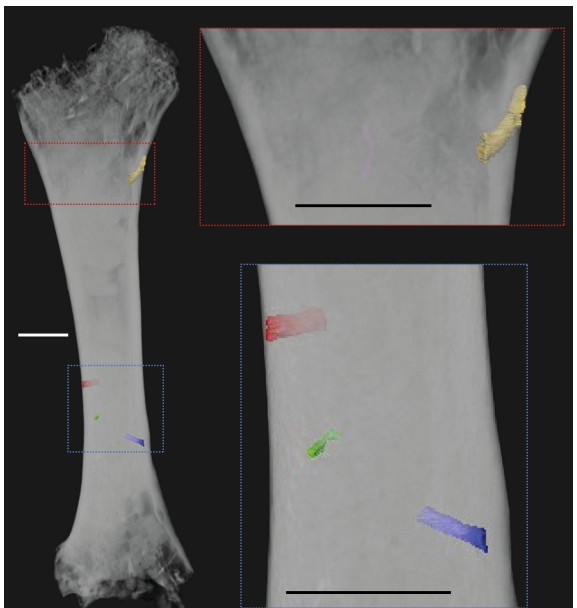

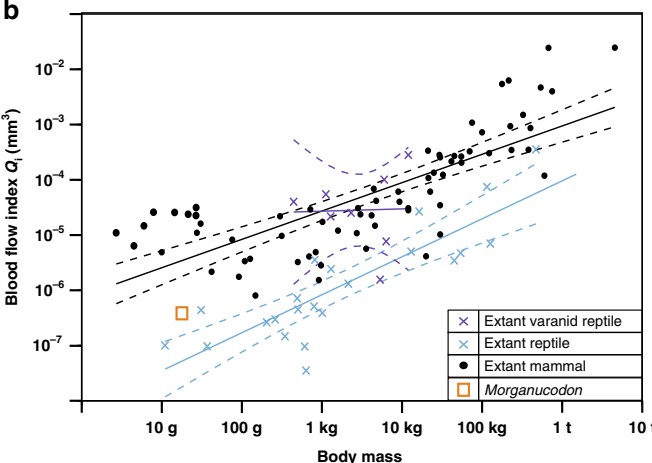

**Fig. 7 Femoral foramina and estimates of relative femoral blood flow for *Morganucodon*. a** 3D reconstruction of *Morganucodon* femur using µCT (specimen UMZC EoPC 19_6, voxel size of 4 µm), with all identifiable foramina segmented and highlighted (red, green, gold and blue). **b** Log$_{10}$ biplot of mean body mass (g) against estimated blood flow index ($Q_i$; mm$^3$) for extant non-avian reptiles ($n = 22$), extant varanid reptiles ($n = 8$), extant mammals ($n = 69$) and *Morganucodon*. GLS regression lines in **b** are shown for extant mammals (black), extant non-varanid reptiles (blue) and extant varanids (purple), with 95% confidence intervals represented by dashed lines. Scale bars in **a** represent 1 mm. Source data are provided as a Source data file.

flow estimates ($Q_i$) suggest that the MMR of *Morganucodon* was intermediate between extant non-varanid reptiles and mammals. We therefore infer that in *Morganucodon* increased MMR (and so also absolute aerobic capacity (AAC) = MMR – BMR) was initially selected for before BMR and that the MMR-first hypothesis[6] is the best-supported model for the evolution of mammalian endothermy. We suggest that at least *Morganucodon*, if not also *Kuehneotherium*, occupied a metabolic grade approaching extant varanids: able to undergo longer bouts of aerobically demanding activity than non-varanid reptiles but not capable of sustaining either mammalian levels of aerobic activity or the elevated thermometabolism exhibited by living endotherms.

Evidence from non-mammalian synapsids (including changes in gait[7], long bone histology[15] and development of secondary osteological features correlated with increased metabolic rate[17,18]) indicate unquestionable changes in physiology from pelycosaur- to mammaliaform-grade taxa. Determinate growth[47] and reduction of dental replacement (diphyodonty) in basal mammaliaforms permitted more precise occlusion[52], which has been considered a key innovation in the development of mammalian endothermy by enabling increased assimilation and higher metabolism[53]. However, determinate growth and diphyodonty appear to have preceded the appearance of modern mammalian levels of endothermy, at least in *Morganucodon* and *Kuehneotherium*. We therefore suggest that the development of precise occlusion in basal mammaliaforms[52] may be more associated with dietary specialization and niche partitioning[1].

Comparison of our results to those of other recent studies of physiology in fossil synapsids supports the hypothesis of a complex, mosaic pattern for the evolution of endothermy, with different characters being selected for at different rates through time, and with respect to phylogeny. For example, the size diminution associated with the cynodont–mammaliaform transition[54] may have reversed the evolutionary trajectory of some previous histological proxies for endothermy[55], contributing to the complex, contradictory patterns observed. Our study also suggests that more work is needed to compare fossil and extant ectothermic and endothermic taxa directly in order to better understand their relative metabolic properties. Many previous studies rely on simple binary divisions, such as the presence/absence of fibrolamellar bone and/or respiratory nasal turbinates. These proxies cannot represent accurately the complex series of physiological characteristics that range between "ectothermy" and "endothermy" and are frequently distributed homoplastically across the synapsid phylogeny, individually and with respect to each other. Other studies provide relative data such as preserved apatite oxygen isotopes[24] that allow comparisons with co-habiting ectothermic taxa but cannot be directly compared to extant data and so do not suggest where the studied fossil taxa fall in the metabolic spectrum of extant vertebrates. However, our results are compatible with recent work on living mammals, suggesting that the BMR of the Middle Jurassic (∼170 Ma) mammalian MRCA was comparable to present-day values[13]. This indicates that evolution towards modern-day mammalian endothermy occurred during the 25 million year-long Early Jurassic and suggests that the mammalian mid-Jurassic adaptive radiation[4,5] was driven by this or vice versa.

In conclusion, our data offer a direct link to measurable aspects of endothermy, such as BMR and MMR, at a key point in mammalian evolution. Further work applying these methods to additional Mesozoic mammaliaforms and mammals, and comparison with evidence from other physiological characteristics, will allow the evolutionary tempo and mode of multiple aspects of mammalian physiology to be determined. The early mammaliaforms *Morganucodon* and *Kuehneotherium* possessed surprisingly low, reptile-like metabolic rates plus a mixture of plesiomorphic and derived characters[7] relating to life history and physiology. Ultimately, we can no longer assume that the endothermic metabolism of living mammals had evolved in the earliest mammaliaforms.

## Methods

**Choice of fossil specimens**. All *Morganucodon* and *Kuehneotherium* specimens are from the Early Jurassic St Brides Island fissure suite, from Glamorgan, South Wales (UK)[1,56,57]. *Morganucodon* specimens used for the cementum analysis are from the collections of the Natural History Museum, London, UK (NHMUK), *Kuehneotherium* specimens are from the collections of the NHMUK and University Museum of Zoology Cambridge, UK (UMZC) and the *Morganucodon* femora specimens are from the collections of UMZC. The *Morganucodon*

specimens are all one species, *M. watsoni*, from Pontalun 3 fissure, excavated in 1962–1963 in Pontalun quarry (now known as Lithalun)[45,56]. *Kuehneotherium* is not abundant in Pontalun 3 fissure and the specimens used here for cementum analysis are from three Glamorgan fissures in Pontalun and Pant quarries: Pontalun 3, Pant 2, and Pant 4. Pontalun 3 and Pant 2 fissures have a relatively impoverished fauna, and Pant 4 has a more diverse biota[45] (Supplementary Data 2). *Kuehneotherium* from Pontalun 3 fissure is *Kuehneotherium praecursoris*[57], but those from the Pant fissures are considered to be different species, based on small molar differences[58]. All specimens from Pontalun 3 fissure in the NHMUK (*Morganucodon* and *Kuehneotherium*) were prepared by immersing dried blocks of clay matrix in hot tap water, with the addition of dilute hydrogen peroxide or sodium hexametaphosphate (Calgon) if required, but those in the UMZC collection were prepared with 10% acetic acid. *Kuehneotherium* specimens from Pant quarry (Pant 2 and Pant 4) were collected by the Kermack team from University College London, between 1955 and 1978, and are from a harder matrix, which was prepared with 15% acetic acid[58].

The *Morganucodon* and *Kuehneotherium* specimens used for cementum analysis are either isolated teeth or dentary specimens with a range of teeth in situ. The lower dental formula of *Morganucodon* is i4:c1:p4:m4–5 and for *Kuehneotherium* it is i?:c1:p6:m6[1]. In *Morganucodon*, the majority of isolated teeth measured were lower second molars (m2) since they are easily identified, are relatively large, have robust separated roots and, as anterior diphyodont molars, they should erupt relatively early and are not replaced, therefore offering a near-complete record of life history. However, i4, c, p3, p4, m1, m3 and m4 teeth were also studied in dentulous *Morganucodon* specimens. Dentulous *Kuehneotherium* specimens are extremely rare and so isolated teeth were scanned in almost all cases. For *Kuehneotherium*, all appropriate teeth were chosen; the distinctive p5 and p6, and a range of lower and upper molars, which can be identified to anterior, mid or posterior tooth row on the degree of triangulation[58].

**Tomographic imaging of cementum**. Pilot scans of two *Morganucodon* lower second molars (NHMUK PV M 104131 and NHMUK PV M 104132) were carried out in 2011 on the nanotomography imaging beamline ID22 at the European Synchrotron Radiation Facility (ESRF), Grenoble, France (project EC 1064). For ID22, we used the following experimental settings for computed tomographic (CT) imaging: X-ray energy of 29.6 keV, 1999 projections over a 180° rotation, 0.5 ms exposure time, 321 nm voxel size, 405 mm sample-to-detector distance, diamond window, a 20-μm-thick LSO scintillator doped with Tb, and 1.5 mm Al filter, single propagation distance tomography.

During a 4-day experiment at the ID19 beamline of the ESRF (18/04/2014-22/04/2014), 71 additional *Morganucodon* specimens (52 isolated teeth and 19 dentaries) and 2 pilot *Kuehneotherium* specimens were scanned (project ES 152). A single harmonic U13 undulator was used as the X-ray source, delivering a pink X-ray beam with peak energy at 26.5 keV, with a 1.4 mm Al filter used to cut the background lower energies. The detector was a microscope optic system coupled to a sCMOS sensor (PCO edge 5.5), mounted with a 10-μm thick GGG:Eu scintillator. Scans were performed using single propagation distance tomography (15 mm sample-to-detector propagation distance), an exposure time of 250–300 ms, 2499 angular projections over a 360° scan and at voxel sizes of 280, 347 and 700 nm.

Subsequently, 117 additional *Kuehneotherium* specimens (116 isolated teeth and one dentary) and 12 additional *Morganucodon* dentary specimens were scanned during a 3-day experiment at the TOMCAT tomographic beamline of the Swiss light Source (SLS), Villigen, Switzerland (13/04/2015-16/04/2015). The beam was set at an energy of 20 keV using a double multilayer monochromator, a LSO:Tb scintillator and a pco.EDGE 5.5 detector. Samples were scanned using single propagation distance tomography (14 mm sample-to-detector propagation distance), an exposure time of 150 ms and 1500 angular projections over a 180° scan at a voxel size of 330 nm. A *Kuehneotherium* lower molar (UMZC Sy 141) was imaged at 1.2 μm voxel size (with an exposure time of 150 ms and 1500 angular projections over 180°) to provide the 3D volume presented in Fig. 1b.

Three juvenile *Morganucodon* dentary specimens, with roots from final deciduous premolars (NHMUK PV M 27312, NHMUK PV M 27474 and NHMUK PV M 27475), and an older individual with extensive molar wear (NHMUK PV M 27465), were scanned during a 3-day experiment at the TOMCAT beamline of the SLS (07/03/2016-10/03/2016). The beam energy was set at 21 keV using a double multi-layer monochromator. Samples were scanned using single propagation distance tomography (14 mm sample-to-detector propagation distance), an exposure time of 200 ms and 1601 angular projections over a 180° scan at a voxel size of 330 nm.

CT reconstructions of the above tomographic data were generated using a filtered back-projection algorithm coupled with "Paganin-style" single distance phase retrieval[59] algorithms developed in-house at the respective beamlines[60,61]. For data from ID19, $\beta = 8.1 \times 10^{-8}$, $\delta = 9.8 \times 10^{-9}$. For data from TOMCAT, $\beta = 3.7 \times 10^{-8}$, $\delta = 1.7 \times 10^{-10}$.

Of the 71 *Morganucodon* molar specimens imaged at beamline ID19 in 2014, 4 were additionally imaged at the nano-imaging beamline ID16A of the ESRF synchrotron (project ES 152). These were imaged using holotomography[62] from four propagation distances. Holograms were recorded using a charge-coupled device detector with an effective pixel size of 3 μm and a 23-μm-thick GGG:Eu scintillator at both 17 keV and 33.6 keV. The selected voxel sizes were 10, 25, 30

and 130 nm. The number of angular projections recorded over 180° varied between 1200 and 2000 and the exposure times were set at 250–800 ms. To generate the image containing the virtual section in Fig. 1g with 30 nm voxel size, the four focus-to-sample distances were 2.65, 13.19, 15.36 and 19.87 mm and the sample-to-detector distance was 1.2648 m.

To produce the 3D model presented in Fig. 1a, a *Morganucodon* lower molar (NHMUK PV M 104134) was imaged using μCT at the University of Helsinki in March 2013. μCT was performed using a Nanotom 180 NF (phoenix X-ray Systems & Services GmbH) with a CMOS detector (Hamamatsu Photonics) and a high-power transmission-type X-ray nanofocus source with a tungsten anode. A total of 900 angular projections were collected for a 180° rotation, at an exposure time of 1840 ms and a voxel size of $2 \times 2 \times 2 \, \mu m^3$. The raw projection data were reconstructed using filtered back-projection by the reconstruction software datos|x rec, supplied by the system manufacturer.

**Increment counting and creation of virtual thin sections.** Cementum increments were counted in CT data using modifications to the techniques suggested by the Cementochronology Research Program[38] to take into account the 3D nature of the PPC-SRμCT cementum data. First, the cementum was visually inspected throughout the entire volume of each scan, in transverse PPC-SRμCT slices using ImageJ/Fiji[63] to distinguish between specimens that could be confidently interpreted as preserving cementum increments or those that were too badly affected by diagenesis for increment counting. Phase-contrast imaging of incremental features is understood to be prone to recurrent destructive interference patterns from Fresnel diffraction that create periodic blurring at differing frequencies when they are scanned using inappropriate experimental parameters (principally X-ray energy, sample-to-detector propagation distance and voxel size). However, our parameters produce blurring frequencies that are too narrow (approximately 500–900 nm) to significantly affect the contrast between cementum increments (1–3 μm radial thickness) (Tafforeau, personal observation). Therefore, no significant masking of increments from Fresnel diffraction blurring should be expected in our data. In specimens that preserved increments, volumes were inspected by eye to identify regions of highest increment contrast with no lensing and/or coalescence between increments. Increments identified in these regions were followed by eye throughout the entire cementum tissue surrounding these regions, both longitudinally and transversely through the root, in order to distinguish between principal increments and accessory increments formed by lensing and coalescence of primary increments in discrete portions of the tissue (Supplementary Fig. 1). Primary increments were distinguished as those that persisted vertically through the entire scanned region of cementum, whereas accessory increments lasted only for short periods before coalescing into neighbouring increments (Supplementary Fig. 1).

Once regions of highly contrasting primary cementum increments had been identified, virtual thin sections of these regions were created. This was performed by isolating ten transverse PPC-SRμCT slices through each region and summing their greyscale values using the "Sum slices" option of the "Z projection" tool in ImageJ/Fiji to create a new image of increased contrast between dark and light cementum increments and reduced image noise. Between three and five virtual thin sections were created for all specimens with readable cementum increments. For each virtual thin section, increments were counted manually by three different observers: Observer One (E.N.) had considerable experience in counting cementum increments (>100 specimens studied); Observer Two (K.W.) had training in counting cementum increments (30 specimens studied under guidance from Observer One) and experience in studying growth patterns in PPC-SRμCT data of long-bones; Observer Three (C.N.) had no prior experience in counting increments or studying growth patterns. Each observer studied virtual thin sections blind, after collections of virtual thin sections were numbered and randomized between specimens using the RAND function in Microsoft Excel. For each observer, the final increment count for every specimen was determined as the maximum number counted in all of its virtual thin sections.

Once each observer had counted increments in every virtual thin section (Supplementary Fig. 3 and (Supplementary Data 1) and the precision between their increment counts was compared by calculating the CV (Eq. 1; Supplementary Table 1):

$$CV = \left[\frac{\text{standard deviation}}{\text{mean}}\right]^{*} 100 \qquad (1)$$

**3D modelling of cementum increments.** 3D modelling was performed, using the Avizo image analysis software (Avizo 8.0; Thermo Fisher Scientific), on a sub-sample of teeth comprising the first tooth imaged using PPC-SRμCT (NHMUK PV M 104131) and the specimens that provided the highest cementum increment counts for each fossil taxon (NHMUK M 104127 for *Morganucodon* and UMCZ Sy 141 for *Kuehneotherium*). Original tomographic data were downsampled in each axis by a factor of two to decrease manual processing time while retaining sufficient spatial resolution to preserve the cementum increments. Principal increments originally identified by eye were manually traced in each PPC-SRμCT slice and assigned to different materials in the "label field" tool-kit. Models were subsequently created and analysed using the "surface view" feature of Avizo. This allowed the pattern of incrementation to be viewed with a 3D perspective, to test

whether increments defined in two-dimensional slices were true principal increments or diffuse accessory increments (Supplementary Fig. 1).

**Fossil body mass estimation and choice of extant taxa.** Body mass for our fossil taxa was estimated using two techniques, based on scaling between single cranial dimensions and measured body mass in extant mammals. Maximum body mass estimates were made using the scaling relationship (Eq. 2) between dentary length (mm) and body mass (g) published for extant marsupial mammals of small body mass and subsequently used to estimate body mass in several Mesozoic fossil taxa from the Late Jurassic Morrison Formation and elsewhere[48]:

$$\ln \text{body mass(g)} = 2.9677^{*}(\ln \text{dentary length(mm)}) - 5.6712 \qquad (2)$$

Dentary lengths are from published CT reconstructions[1] (20 mm for *Morganucodon* and 21.9 mm for *Kuehneotherium*), resulting in body mass estimates of 25.0 g and 32.7 g respectively. However, this estimate may be an overestimate for *Kuehneotherium* as it has a longer, more gracile, dentary relative to *Morganucodon*, due to the differing feeding ecologies of the two taxa[1].

Second, minimum body mass estimates were calculated using the scaling relationship (Eq. 3) between skull length (mm) and body mass (g) found for 64 extant species of small "lipotyphlan" insectivores[64] and used to estimate the body masses of several Mesozoic mammaliaforms[21,64,65].

$$\ln(\text{body mass(g)}) = 3.68^{*}(\ln(\text{skull length(mm)})) - 3.83 \qquad (3)$$

Due to a lack of complete, diagnostic cranial material for UK samples of *Morganucodon* and *Kuehneotherium*, their skull lengths were estimated using a scaling relationship between dentary length and skull length calculated for *Morganucodon oehleri* from Rowe et al. Figure S1[21] (skull length (mm) = 1.0458 × jaw length (mm)). Following this relationship, we used dentary length[1,21] to estimate a skull length of 21.0 mm for *M. watsoni* and 22.9 mm for *Kuehneotherium*, which results in body mass estimates of 10.7 and 14.9 g, respectively (as above, this may be an overestimate for *Kuehneotherium*). Body masses presented in Figs. 5 and 7 are mean values of the two estimates (17.9 g for *Morganucodon* and 23.8 g for *Kuehneotherium*).

**Extant data.** Information on maximum wild lifespan and mean body mass was obtained for 278 extant terrestrial mammal species (body mass: mean = 70.2 kg, standard deviation = 395 kg; wild lifespan: mean = 10.8 years, standard deviation = 11.2 years) and 256 extant terrestrial non-avian reptile species (body mass: mean = 3.6 kg, standard deviation = 20 kg; wild lifespan: mean = 12 years, standard deviation = 14 years) (Supplementary Data File 3). Information on maximum captive lifespan was obtained for 644 extant terrestrial mammal species (body mass: mean = 62.2 kg, standard deviation = 324 kg; captive lifespan: mean = 15.5 years, standard deviation = 11.5 years) and 866 extant reptile species (body mass: mean = 2.4 kg, standard deviation = 15.9 kg; captive lifespan: mean = 15.3 years, standard deviation = 11.6 years). Information for flying or gliding taxa (including birds) was not included as their body masses are known to be secondarily reduced to aid their lifestyle and so may distort observed overall relationships between lifespan and body mass. In particular, birds and bats live on average three-to-four times as long as terrestrial mammals of similar body mass[66]. Similarly, marine mammals were not included as their environment positively affects body mass, allowing significantly higher body masses than terrestrial taxa[67]. Mammalian wild and captive lifespan data were obtained from the primary literature. The majority of these data originated from a download from the online databases of the Max Planck Institute (https://www.demogr.mpg.de/longevityrecords/0203.htm) on 10/03/2019. The majority of reptile wild and captive lifespans were obtained from the supplementary information of Scharf et al.[68]. Body mass estimates were obtained from an online *Ecological Archives* database[69] (http://www.esapubs.org/archive/ecol/E084/094/metadata.htm), the AnAge database (https://genomics.senescence.info/species/) and published data[50,68]. Although the Max Planck Institute data are entitled "longevity records", it notes in the introduction that this is equivalent to "highest documented age". The authors of the database believe that, despite any analytical and conceptual shortcomings, the records are useful in a number of comparative and disciplinary contexts, including demographic, gerontological, ecological and evolutionary. Similarly, the AnAge database states that maximum longevity is synonymous with maximum lifespan.

Data on SMR for extant mammals (n = 117; mean = 915; standard deviation = 3018) and reptiles (n = 55; mean = 62.3; standard deviation = 159) was obtained from a published electronic appendix[50]. SMRs for mammals were included in this data set only if measured under basal metabolic conditions[50] and so are synonymous with BMRs.

Data on the postnatal growth rate constant K was obtained upon request for extant mammals[32] (n = 115, mean = 0.03, standard deviation = 0.03) and reptiles (n = 33, mean = 0.002, standard deviation = 0.003) from previous publications[70,71]. The growth rate constant K is a magnitude free measure of growth rate, measured in days$^{-1}$, that is considered the best measure for comparing growth rates between multiple species[72].

Physiological metrics (lifespan/body mass/SMR/K) were compared using PGLS regression in the "R" statistical environment with the "ape", "geiger", "nlme" and "phytools" packages installed. For each regression, phylogenetic subsets of squamate and mammal taxa were downloaded from https://vertlife.org (100 trees

per subset, birth-death node-dated completed tree distribution for mammals) representing the phylogenetic relationships of every taxon in the respective subsample. For crocodilian taxa, a phylogeny was manually constructed in R following the time-calibrated phylogeny of Oaks[73] (using node mean age values from the species tree/90 My maximum analysis) and added to the base of the squamate phylogeny to produce a reptile clade. Each subset was investigated to find the tree that produced the highest PGLS $F$ values and this tree was then input into the "corPagel" covariance structure for the "gls()" function to produce a phylogenetically informed regression model between the respective metrics.

The relationships between body mass and maximum lifespan were compared between extant mammals and extant reptiles using phylogenetic ANCOVA following Smaers et al.[74] using the "gls.ancova" function in "R" with the same libraries installed as for PGLS. The individual phylogenetic subsets representing the mammal and reptile samples were combined with connecting branches connecting the Last Common Ancestors (LCAs) of the crocodilian, squamate and mammal clades. The LCA node connecting squamates and crocodilians was dated at 275.9 Ma using the mean of minimum and maximum node ages from Benton et al.[75]; similarly, the LCA node connecting reptiles (squamates and crocodiles) and mammals was dated at 325.5 Ma[75]. The mammal and reptile clades were then distinguished using two factor elements, and the influence of these factors upon the fit of the data to a series of models of increasing complexity was tested. First, a baseline model was produced consisting of one slope and one intercept for the entire data set. Three models were then compared to this baseline; the first varied in slope between mammals/reptiles, keeping their intercepts constant; the second allowed the intercept to vary between mammals/reptiles, keeping their slopes constant; and the final model allowed both slope and intercept to vary between mammals/reptiles. Comparisons between these models allowed the determination of their improvements upon the baseline model. Phylogenetic ANCOVA identified a lambda value of <0.001 between body mass and $Q_i$ for reptiles and mammals, and so non-phylogenetically informed GLS and ANCOVA were used to compare these metrics.

### µCT study of femoral nutrient foramina.
Nutrient foramina in the femora of *Morganucodon* and 11 extant mammal taxa of comparable size were imaged using µCT in the X-Ray Micro-Imaging Laboratory, University of Helsinki (Supplementary Table 2). We used µCT data of femoral foramina rather than photographic images of the foramina[51] because many of the *Morganucodon* foramina were filled with sediment, making their photographic measurement difficult or impossible, whereas the differential density of sediment and fossilized bone in µCT data allowed visualization and analysis. Six *Morganucodon* femur specimens were selected for analysis from the collections of the Cambridge University Museum of Zoology based on their relative completeness. All femora (both fossil and extant) were scanned using a Bruker Skyscan 1272 µCT scanner at 70 kVp source tension and with a 0.5 mm Al filter. For each scan, 1125–1800 angular projections were collected during a 180° rotation, at an exposure time of 1344 ms. One fossil specimen was scanned at an isotropic voxel size of 4 µm (Fig. 7a) and five at 5 µm. All femora from extant taxa were scanned at 4 µm voxel sizes. The raw projection data were reconstructed using filtered back-projection with the Feldkamp algorithm by the Bruker reconstruction software "NRecon" (Version 1.7.1.0).

Analysis of µCT femora data was conducted using the Avizo image analysis software (Version 9.3.0; Thermo Fisher Scientific). Foramina were located using the "Orthoslice" tool, to scroll through transverse CT slices of each femur. Once located, the foramen was imaged in 3D using the "Volume rendering" tool. This allowed the minimum diameter, and so minimum radius (in cm), of the foramen aperture to be assessed and measured using the "3D line measurement" tool. Minimum diameter was used by Seymour et al.[51], in the study of foramina in small mammals, as it is not always possible to see the direction of nutrient vessel penetration. Resulting foramina radii were used to generate estimates of blood flow index ($Q_i$) following the method outlined in Seymour et al.[51]. If multiple foramina were found in a single specimen, then the radii were summed following the assumption that this represents the total entry/exit potential of nutrient circulation through the femur[51]. This sum radius, or the single radius in specimens with single foramina (both abbreviated to $r$; measured in cm), was then used along with the length of the femur ($L$; measured in cm) to estimate $Q_i$ as follows (Eq. 4):

$$Q_i = r^4/L \qquad (4)$$

Although *Morganucodon* femoral elements are relatively common in the *Hirmeriella* fissure suite, almost all are incomplete[76]. We here chose the six most complete femoral specimens known from the Pontalun 3 fissure, from where the *Morganucodon* teeth imaged using PPC-SRµCT originated. Specimens chosen (UMZC EoPC 19_1 to EoPC 19_6) preserve at least two-thirds of the femoral shaft. Length was estimated from the minimum mid-shaft diaphysial width of each element, using the scaling relationship between the two measurements established for a reconstructed *Morganucodon* femur created by concatenating three incomplete femora by P.G.G. (Eq. 5):

$$\text{Femur length} = 10.3^* \text{ maximum mid-diaphysial diameter.} \qquad (5)$$

### Statistics.
One-way ANOVA comparison of intra-observer CV between cementochronological studies; Shapiro–Wilk normality test: PPC-SRµCT data

$W = 1$, $p = 1$, histological data $W = 0.93$, $p = 0.41$; test statistics $F = 11.12$, degrees of freedom (df) = 10, Cohen's effect size $d = 3.13$, $p = 0.00728$. Phylogenetic ANCOVA comparison of PGLS regression slopes for lifespan against body mass in mammals ($\log_{10}$ lifespan = $0.26(\log_{10}$ body mass$) + 0.16$; 95% confidence interval (CI) = 0.05; $r^2 = 0.69$) and reptiles ($\log_{10}$ lifespan = 0.26 ($\log_{10}$ body mass$) + 0.60$; 95% CI = 0.08; $r^2 = 0.46$); slopes are statistically similar ($F = 0.868$, $p = 0.352$) while means are significantly separated ($F = 4.44$, df = 529, partial eta squared effect size = 0.32, $p = 0.036$). PGLS regression of mammalian lifespan against msSMR: $\log_{10}$ msSMR = $-0.237(\log_{10}$ lifespan$) - 0.083$; 95% CI = 0.07; $r^2 = 0.59$, $p < 0.001$. PGLS regression of reptilian lifespan against msSMR: $\log_{10}$ msSMR = $-0.83(\log_{10}$ lifespan$) - 0.31$; 95% CI = 0.255; $r^2 = 0.43$, $p < 0.01$. PGLS regression of mammalian lifespan against growth constant $K$; $\log_{10} K = -0.692(\log_{10}$ lifespan$) - 1.171$; 95% CI = 0.101; $r^2 = 0.66$, $p < 0.01$. PGLS regression of reptilian lifespan against growth constant $K$; $\log_{10} K = -0.69(\log_{10}$ lifespan$) - 2.523$; 95% CI = 0.339; $r^2 = 0.43$, $p < 0.01$. One-way ANCOVA comparison of OLS regression slopes for $Q_i$ against body mass in extant mammals ($\log_{10}(Q_i) = 0.513 \times \log_{10}($body mass$) - 6.104$) and non-varanid reptiles ($\log_{10}(Q_i) = 0.685 \times \log_{10}($body mass$) - 8.139$); slopes are statistically similar ($F = 2$, $p = 0.16$) while means are significantly different ($F = 87.6$, df = 89, partial eta squared effect size = 0.50, $p = 7.4E^{-15}$).

**Reporting summary**. Further information on research design is available in the Nature Research Reporting Summary linked to this article.

### Data availability
The tomographic data that support the findings of this study are available from the corresponding authors upon reasonable request. Virtual thin sections used for cementum and bone increment counts are available at the University of Southampton's PURE data repository as data number D1506 (https://doi.org/10.5258/SOTON/D1506). Physiological and phylogenetic data are from online databases of the Max Planck Institute (https://www.demogr.mpg.de/longevityrecords/0203.htm), an online *Ecological Archives* database (http://www.esapubs.org/archive/ecol/E084/094/metadata.htm), the AnAge database (https://genomics.senescence.info/species/), the VertLife online project (https://vertlife.org) and the literature (references in Supplementary Data file 3) and are provided in Supplementary Tables, as Supplementary Data files, and as a part of the Source data provided with this paper.

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

## Acknowledgements

We acknowledge the European Synchrotron Radiation Facility, Grenoble, France for provision of synchrotron radiation facilities on beamlines ID19 and ID16A (project ES152) and thank Peter Cloetens for assistance in using beamline ID16A. We also acknowledge the Paul Scherrer Institut, Villigen, Switzerland for provision of synchrotron radiation beamtime at beamline TOMCAT of the Swiss Light Source (project 20141278). The research leading to these results has received funding from the European Community's Seventh Framework Programme (FP7/2007-2013) under grant agreement no. 312284 (for CALIPSO). We thank Keijo Hämäläinen for his help in the initial stages of the synchrotron imaging. This project was part-funded by a Natural Environmental Research Council studentship and an Engineering and Physical Sciences Research Council studentship, awarded to E.N. and P.S. (Grant number NE/R009783/1), and we also thank the Academy of Finland for part-funding the project. Thank you to the Natural History Museum London for contributing to travel for P.B. via the Departmental Investment Fund for Earth Sciences. We thank Ginko Investments Ltd. for funding for materials and travel and the University of Bristol Bob Savage memorial fund for travel for E.N. Many thanks to the Natural History Museum, London, University Museum of Zoology, Cambridge and the Finnish Museum of Natural History, Helsinki, Finland for loans of specimens. facilitated by Martha Richter, Rob Asher, Matt Lowe and Martti Hildén. For assistance with laboratory work and materials, we thank Tom Davies, Wendy Dirks, Dani Schmidt, Remmert Schouten, Pedro Viegas, John Cunningham & Duncan Murdock. Discussions: Roger Benson, Chris Dean, Wendy Dirks, Jim Hopson, Fabien Lafuma, Thomas Martin, Rachel O'Meara, Stephen Naji, Tanya Smith and Emily Rayfield.

## Author contributions

I.J.C. and P.G.G. conceived and designed the project. E.N., P.G.G., P.B., K.R. and I.J.C. selected, prepared and curated specimens. E.N., P.G.G., P.B., V.F., D.H., T.K., A.K., A.P., P.S., H.S., P.T., B.Z.-P. and I.J.C. performed the synchrotron experiments. E.N., A.K. and I.J.C. performed the microCT experiments. E.N. processed and E.N., C.N. and K.W. analysed the synchrotron data. E.N. and I.J.C. analysed the micro-CT data. E.N., P.G.G. and I.J.C. discussed the interpretations. E.N. wrote the first draft and created all figures; E.N., P.G.G. and I.J.C. wrote the manuscript; all authors provided a critical review of the manuscript and approved the final draft. Authors M.J.B., V.F., N.J.G., D.H., J.J., T.K., A.K., C.N., A.P., K.R., K.R.B., P.S., H.S., P.T., K.W. and B.Z.-P. contributed equally to this work and are listed in alphabetical order. This article originated as a Master's thesis (University of Bristol), then a PhD thesis (University of Southampton), performed by E.N. and supported and supervised by P.G.G., P.S., N.J.G., J.J., K.R.B., M.J.B. and I.J.C.

## Competing interests

The authors declare no competing interests.
