## [Peer Review File · Nature Communications]

Reviewers' Comments:

Reviewer #1:

Remarks to the Author:

Ms entitled "Reptile-like physiology in Early Jurassic stem-mammals" is a landmark contribution to our knowledge of the origin of mammalian thermophysiology. Authors quantify incremental tooth cementum using highly sophisticated synchrotron X-ray tomographic imaging methods to infer lifespan and basal metabolic rate of Early Jurassic Morganucodon and Kuehneotherium. Moreover, they infer maximum metabolic rate of Morganucodon using femoral nutrient foramina size.

In strong contrast with the sophisticated methods used to quantify measurements, authors use statistical methods that do not take into account the phylogeny. We know ever since the seminal publication by Joseph Felsenstein "Phylogenies and the comparative method" 34 years ago in *The American Naturalist* that interspecific studies that do not include the phylogeny obtain flawed results. Hundreds of papers have been published on this topic. It is surprising to see that Ordinary least squares regressions (e.g., l. 175) and ANCOVAs (e.g., l. 182) were performed without including the phylogeny, thus ignoring progresses made during more than three decades in this field. I suggest to repeat analyses using methods that incorporate the phylogeny: Phylogenetic Generalized Least Squares and Phylogenetic ANCOVA (please see the papers below)

Another aspect deserves attention: the variable mass-specific standard metabolic rate needs to be standardized. Considering that mass-specific standard metabolic rate strongly depends on the surface/volume ratio, a useful standardization is that used by, for instance, Seymour et al (2014): $\text{mLO}_2/\text{h} \cdot \text{g}^{-0.67}$

References

Martins, Emilia P.; Hansen, Thomas F. (1997).

"Phylogenies and the Comparative Method: A General Approach to Incorporating Phylogenetic Information into the Analysis of Interspecific Data".

The American Naturalist. 149 (4): 646–667.

Fuentes-G JA, Housworth EA, Weber A, Martins EP. (2016)

« Phylogenetic ANCOVA: Estimating Changes in Evolutionary Rates as Well as Relationships between Traits. »

The American Naturalist 188 (6) : 615-627.

Seymour R.S., Bennett-Stamper C.L., Johnston S.D., Carrier D.R., Grigg G.C. (2004).

Evidence for endothermic ancestors of crocodiles at the stem of archosaur evolution. *Physiol. Biochem. Zool.* 77:1051–1067.

Reviewer #2:

Remarks to the Author:

Review of Newham et al "Reptile-like physiology in Early Jurassic stem-mammals" for *Nature Communications*

Zhe-Xi Luo

The University of Chicago

Email: zxluo@uchicago.edu

Appraisal

The paper by Newham and colleagues is highlighting an important question in evolution of mammals. It represents a major progress in the understanding of the life-history characteristics for early mammals.

Extant mammals differ from extant nonmammalian vertebrates in their life-history characters, especially in growth pattern and metabolic rates. This study is aimed to estimate the lifespan – an important life history trait - from the tooth root cementum of the early mammaliaforms *Morganucodon* and *Kuehneotherium*. Early mammaliaforms are already known to be modern mammal-like in many (but not all) of their characters. As lifespans are correlated with growth model and with metabolism rates in modern mammals, and if lifespan can be established for the early mammaliaforms by inference from the cementum increments of fossil teeth, it can help to reveal further insight about the life-history characteristics and the physiological function of these early mammals.

To the extent that lifespans can be established for *Morganucodon* and *Kuehneotherium* by first hand observation and with statistical robustness, this study is a giant step forward for characterizing the yearly age and for estimates of metabolic rates for mammaliaforms, which are intermediate between extant mammals, and the pre-mammaliaform cynodonts.

This paper has a strong research merit – Newham and colleagues have accomplished a sophisticated study by high resolution of synchrotron scans to examine the tooth root cementum increments (“growth layers”) of two of the earliest-known mammaliaforms. They have also sampled some impressively large numbers of specimens of *Morganucodon* (36 specimens with discernible growth layers out of 87 specimens examined) and of *Kuehneotherium* (27 specimens showing growth layers out of 119 specimens examined). Following the best practice, the cementum increments are independently counted by three different observers, with robust statistics on the confidence level of the first-hand observation.

The authors provided some important and very valuable benchmark data:

Morganucodon:

- The longest living specimen or “Longevity champion” is aged to 14 years;
- the mean age at 4.78 years of the entire cohort (the mean of 36 specimens reported in Extended-Data Table 1)

“maximum lifespan” can be calculated at 10 years for *Morganucodon*

Note: The standardized definition of maximum lifespan for demographics studies and biology of senescence is the mean lifespan of the most long-lived 10% of a given cohort. Among the 36 *Morganucodon* specimens, the 4 specimens of the highest cementum increments are 14, 10, 8 and 8 increments (“years”).

Kuehneotherium:

- The longest living specimen or the “Longevity champion” is aged at 9 years
- the mean age of 4.98 years (calculated as mean of 36 specimens reported in Extended-Data Table 1)

“maximum lifespan” can be calculated at 8.7 years for *Kuehneotherium*

Note again: the standard definition for maximum lifespan for demographics and biology of longevity is “the mean lifespan of the most long-lived 10% of a given cohort.” Based on the *Kuehneotherium*

specimens with the three highest cementum increments of 9, 8 and 9, out of the 27 Kuehneotherium specimens (Extended Data Table 1).

Synchrotron scans are expensive and the visualizing the data from these scans are extremely time-consuming. The resulted datasets for this study are an enormous accomplishment by itself. I like to compliment Newham and colleagues for their technical skills and time-effort in producing these valuable datasets.

I also found it informative and refreshing that Newham and colleagues used the scaled vascular foramina diameters/volumes of femora as proxies to the blood supply. These are proxies to maximum metabolic rate (MMR) that can be applied to Morganucodon. I found this part of the comparative analysis well reasoned, and technically well justified. The key result is also very reasonable: Morganucodon's MMR is now estimated to be intermediate between the MMR's of extant mammals and the MMR's extant reptiles. Given the intermediate phylogenetic positions of Morganucodon and Kuehneotherium in cynodonts-mammaliaforms-mammals evolution – this makes sense. Moreover, this result has revealed a new physiological understanding that was not previously available.

The scaling analyses of yearly lifespans of both fossil mammaliaforms and the reference extant mammals with the basal metabolic rate (BMR) and the growth rate constant K (Extended Data Figure 4) show that there are some consistent patterns of these metrics – the estimated growth rate and bmr of mammaliaforms are essentially just like those of extant mammals – if based on lifespans (increments).

However, if scaling the lifespan (years) with the body mass (main text Fig. 4), the story would be different – here the two mammaliaforms are quite different from extant mammals, but more similar to extant reptiles.

A Major Revision

I think that Newham and colleagues incorrectly interpreted the meaning (definition) of maximum lifespan vs. wildlife lifespan. I offer three inter-related comments and suggestions for revision:

Suggestion 1- Newham and colleagues defined the maximum lifespan here on a singular "long-life champion" specimen with 14 increments for Morganucodon, out of the 36 specimens of this Morganucodon sample. However, the maximum lifespan is defined differently in demographic and senescence studies - In demographics, and in senescence and longevity-related biological studies, the "maximum lifespan" is defined as the mean lifespan of the most long-lived 10% of a given cohort, not a "longevity champion age" (or "Guinness World Record") -14 increments in one specimen – as the authors picked for Morganucodon. Given the well documented data across an extensive sample of Morganucodon teeth, the "maximum lifespan" should be recalculated as the mean increment of the top 4 specimens of a sample of 36 specimens.

To pick a "longevity champion" at 14 increments is far less justifiable statistically, as it deviates from the standard of the maximum lifespan in demographic studies of extant mammals (defined as the "longest-10%-mean.") I really urge that the authors follow the "mean of top 10% of longest living individuals of the given cohort", and use the top 4 specimens for the mean of this metric. It so, this change would lead to a revision of 14 increments (longevity champion) to 10 increments (the mean of four specimens with most increments) for Morganucodon. Similarly the 9 increments of Kuehneotherium would be revised to 8.7 increments.

This would be the first step. Then authors should place 10 increments for Morganucodon and 8.7

increments for *Kuehneotherium* in regression analyses (Figure 4B).

Suggestion 2 – After the revision of the maximum lifespan as “the mean of longest-living 10% of the total cohort of the cementum sampling, if Newham and colleagues still prefer to use “maximum lifespan” as a key metric for comparative discussion, then they should also revise and recompose the comparative data of extant mammals. They should use the captivity longevity of extant mammals against body mass of extant mammals in regression (Figure 4B). The reason for this suggestion is that the captivity longevity is a better representation of the realized lifespan, while the maximum lifespan cannot be reliably observed in real wildlife population. The AnAge database (<https://genomics.senescence.info/species/>) is a live and curated database, and the most widely used in demographics and senescence studies. This mainstream database shows the maximum lifespan in captivity as the default maximum age, and the only formally listed yearly age.

Newham and colleagues did not use this default age estimate from AnAge database for the majority of mammals in their reference dataset. Instead, for the wildlife lifespan estimates of the comparative mammal data, they optioned to pick much younger age estimates from the wildlife spans from other studies.

First - If “maximum lifespan” (best-recorded increments of a specimen) is used for *Morganucodon* and *Kuehneotherium*, then for the reference data of extant mammals, the maximum lifespans in captivity should also be used. This is to be biologically consistent.

Second - In demographics of extant mammals, the “longest-documented lifespan in captivity” is the most reliable data, and the most widely available of the various “lifespan metrics.” For the AnAge database (genomics.senescence.info/species/), the longest life span in captivity is the default metric. Further, the AnAge database provides careful comments on the variable nature of “wildlife” lifespan. The data on variation of lifespan (other than known max lifespan by captivity) are discussed in the footnotes, and discussion on reliability ranking is sourced to the literature. The Max Planck database on longevity records have lists of both wildlife lifespan and the captivity longevity, but its taxonomic coverage on longevity in captivity is more extensive than “wildlife lifespan” for the terrestrial (non-cetacean and non-chiropteran) mammals. In sum, the captivity lifespan is the hard data, and more available, and less susceptible to uncertainty.

Third - In many histological studies of cementum growth layers that have traced many individuals across a range of ages for a species, the specimens are mostly (if not entirely) from captive animals, or from animals that were wild-caught, but kept captive. It appears from the tables and graphics of the historical studies by Klevezal and her colleagues that these cementum increment studies are not based on real wildlife populations.

The direct data for the Newham et al. study are the cementum increments. As some of the best available series of cementum increments in extant mammals are sampled from kept populations, it is better to compare the fossil increment data with the extant mammal cementum increments data from captive/kept populations. This is to be consistent in nature of cementum data of both extant mammals, and the fossil species involved.

Suggestion 2 – An alternative way to revise - a better and easier way in my opinion – would be to use the statistical mean of the cementum increments of the 36 *Morganucodon* specimens, which is 4.78 increments. The authors can then plot this number into Figure 4B for the regression of the wildlife lifespan of extant mammals against body mass of extant mammals.

This way, the authors can be biologically consistent parsing their data for comparison, yet they do not

need to go through the tedious and more cumbersome revision of all "wildlife lifespan" data of extant mammals that have already collected and programmed into the current version of the paper. I believe that this alternative is much better for several considerations:

First – the statistical mean (4.78 increments for Morganucodon, 4.98 increments for Kuehneotherium) is more reliable for estimating yearly age, than "maximum" increment (14 for Morganucodon and 9 for Kuehneotherium). Several cementum age estimate papers on large sample sizes have consistently shown that annual c-increments are correlated with yearly age only in young individuals. But the increments can deviate significantly from yearly age in older adults of the same species. Grau et al (1970) study on raccoons (*Procyon lotor*) (ref 73) demonstrated this decline of accuracy of annuli-year correlation in older specimens. Grau (1970) shows that the method works well before 4 years of age, but cementum age estimate is rapidly declining in accuracy for individuals older than 4 years of age. For 5 years or older, cementum rings are only 70%-50% accurate for estimating yearly age. Kay and Cant (1988) (ref 76) on *Maraca mulatta* in a very thorough study - it has demonstrated the correlation of cementum "annuli" with yearly age can vary significantly over the lifespan of individual specimens. In older specimens there is an increasingly wider scattering of datum points on either side of the regression line (Kay and Cant 1988: figure 3) – wider variation (= less accuracy) in older adults that are 12 years or beyond. Further, the older individuals have a tendency to show different annulus counts on the left versus right first molars of the same individual (Kay and Cant 1988: table V).

To put it more simply – when individuals of mammals approach their "maximum lifespan," it is progressively less reliable to count on the cementum "annuli" as a proxy to yearly increments.

For this reason, it is statistically sound and biologically more justifiable if one uses the mean of cementum increments across a sample, such as the 36 specimens counted for their increments for Morganucodon, and 27 specimens for Kuehneotherium in Extended Data Table 1.

Second – the statistical mean of increments of the fossil teeth can be regarded as the peak mortality age of the populations of the two mammaliaforms. This is to take the mean of increment counts as a proxy to the peak age of death. This proxy is relevant to demographics of these fossil taxa in real paleoecological context. Given the large sample size and the fact that teeth of both Morganucodon and Kuehneotherium were collected from multiple quarries/localities, this is a biologically justifiable approach under the time-average principle of paleoecology.

Third - it makes the best sense to compare the peak mortality (mean of the increments across the entire sample) of Morganucodon/Kuehneotherium to the wildlife lifespan data of extant mammals in the reference dataset. The means of these samples are more realistic in terms of paleoecology, and more comparable to extant wildlife mammal lifespan in the context of their ecology. This is better than picking a singular specimen as the symbolic "longevity champion."

Fourth – A great strength of this work is that it has amassed a significant body of data (36 increment counts of Morganucodon and 27 counts of Kuehneotherium). It would be more powerful to fully capitalize these datasets and to use a metric with robust statistics.

I hasten to add – this is a much easier way to revise this manuscript because the statistical means of cementum increments are already available: 4.78 annuli for Morganucodon and 4.98 annuli for Kuehneotherium. There is no need to vet and to revise the entries of raw data from extant mammals, and re-compose the dataset for the regression analysis.

My expectation is that the mean of 4.78 increments Morganucodon and 4.98 for Kuehneotherium are closer to the lifespans of extant mammals, than the previous estimates of 14 for Morganucodon and 9

for Kuehneotherium. Thus the estimates on basal metabolic rate (BMR, or RMR) for these two mammaliaforms would be more likely intermediate between the extant reptiles and extant mammals. This would be more consistent with the estimate on maximum (aerobic) metabolic rate (MMR) from the femoral vascular channels as proxy to blood vessel volumes.

A Clarification

Can the authors clarify a discrepancy that I have noticed between Text Figure 4 and Extended Data Figure S4?

Extended Data Figure S4 shows consistent patterns of Morganucodon/Kuehneotherium and the extant mammals in the standard metabolic rate plus the growth rate constant K of extant mammals, with respect to their cementum-based estimates (fossils) or actual yearly ages (extant). These regression analyses seem to suggest that the growth rate and smr of the two mammaliaforms are in the similar range as those of extant mammals, if scaled according to their ages (=cementum increments for fossils). This (ED Fig. 4) result is sensible.

However, the results are quite different in scaling of the standard metabolic rate and growth rate constant K of Morganucodon/Kuehneotherium and extant mammals with respect to their body masses, as illustrated in text Fig. 4. Here the two mammaliaforms are different from extant mammals, but are more similar to extant reptiles. The main narrative of the paper is based on Fig. 4, but not on Extended Data Fig 4.

Why are placements of Morganucodon and Kuehneotherium are so different in these two sets of regression analyses? I do not disagree with these, but I am puzzled. I suggest that authors provide a discussion to clarify this in Supplementary Information.

Correction on Extended Data Figure 2.

The dicynodonts are a clade. But the four dicynodonts mentioned in this figure – Kawingasaurus, Oudenodon, Lystrosaurus, Moghreberia – are tagged onto the backbone of the phylogenetic tree in four different nodes. This could be misconstrued that the authors would interpret that dicynodonts are paraphyletic (the authors clearly did not intend this) and that these dicynodonts have different phylogenetic positions, with regards to Thrinaxodon. I understand well that this figure is designed to summarize all synapsid taxa that have been inferred for physiological functions, as mapped on geological time scale. But if you use phylogeny at all, you must somehow combine all four dicynodonts into the dicynodont clade, and dicynodonts as a whole should have a single node on the synapsid phylogeny.

Extended Data Figure 2. Typo – For Morganucodon – the “determinant” growth should be “determinate” growth.

Figure 4 – typo – Figure 4d, Y-axis label – Growth rate constant K (days⁻¹). “constant” should be “constant”

Reference 68. The last names of all authors got lost in this listed reference. The name of the journal is also wrong. I think this is Felisa A. Smith, Kathleen S. Lyon, et al. (2003) Ecology.

Summary

I strongly support that Nature Communications should publish this important paper. I just like to

suggest that Newham and colleagues reconsider how they would best parse the comparative data, so that the comparison is biologically justifiable. They should use the mean of increments of the entire sample of mammaliaforms, in comparison with the wildlife lifespan data of extant mammals, as these are ecologically comparable. Or they can continue to use the mean of maximum cementum increments (14 for Morganucodon, 9 for Kuehneotherium), but to do so they would need to re-vamp the comparative extant mammal dataset for the maximum lifespans in captivity.

Reviewers' comments:

Reviewer #1 (Remarks to the Author):

Ms entitled “Reptile-like physiology in Early Jurassic stem-mammals” is a landmark contribution to our knowledge of the origin of mammalian thermophysiology. Authors quantify incremental tooth cementum using highly sophisticated synchrotron X-ray tomographic imaging methods to infer lifespan and basal metabolic rate of Early Jurassic Morganucodon and Kuehneotherium. Moreover, they infer maximum metabolic rate of Morganucodon using femoral nutrient foramina size.

We are grateful for this very positive appreciation of the importance of our work.

In strong contrast with the sophisticated methods used to quantify measurements, authors use statistical methods that do not take into account the phylogeny. We know ever since the seminal publication by Joseph Felsenstein “Phylogenies and the comparative method » 34 years ago in *The American Naturalist* that interspecific studies that do not include the phylogeny obtain flawed results. Hundreds of papers have been published on this topic. It is surprising to see that Ordinary least squares regressions (e.g., l. 175) and ANCOVAS (e.g., l. 182) were performed without including the phylogeny, thus ignoring progresses made during more than three decades in this field. I suggest to repeat analyses using methods that incorporate the phylogeny: Phylogenetic Generalized Least Squares and Phylogenetic ANCOVA (please see the papers below)

We thank the reviewer for their well-considered suggestion and upon review of the cited literature agree that regressions of this nature should be phylogenetically informed. We have thus updated our analysis by using the phylogenetic generalised least squares method (PGLS) to compare our physiological metrics - maximum lifespan/mean body mass/mass-specific standard metabolic rate (msSMR)/post-natal growth rate (K). We also now use phylogenetic ANCOVA to compare the mean body mass/lifespan relationships between our extant samples of mammals and reptiles.

Phylogenetic ANCOVA comparisons of mean body mass/blood flow index (Q_i) relationships between extant mammals and reptiles resulted in a non-significant phylogenetic lambda value (<0.001), and so we retained our non-phylogenetic ANCOVA comparison between these samples. While phylogenetic reconstructions used for PGLS and phylogenetic ANCOVA are likely to be continuously improved, we suggest the new analyses underscore the general robustness of our inferences.

Another aspect deserves attention: the variable mass-specific standard metabolic rate needs to be standardized. Considering that mass-specific standard metabolic rate strongly depends on the surface/volume ratio, a useful standardization is that used by, for instance, Seymour et al (2014): $\text{mLO}_2/\text{h} * \text{g}^{-0.67}$.

The relationship between body size and metabolic rate is a contentious issue in evolutionary and biological science. The question of a universal exponent between logged body mass and logged metabolic rate has been addressed both for specific clades (e.g. mammals) and for all animals and plants (Glazier, 2010). While some workers have claimed that a 0.67 relationship exists for all animals resulting from surface/volume ratios, others suggest a universal $\frac{3}{4}$ power law arises from the geometry of resource-transport networks (Glazier 2015). Studies focussing on just mammals have found that molecular phylogenies suggest a 0.67 scaling exponent, while morphological phylogenies support an exponent of 0.75 (Kleiber’s Law) (Symonds and Elgar, 2002). Uncertainty remains regarding both absolute values, as they have proved to be dependent on the phylogeny used and scaling can vary significantly between closely related taxa dependent on lifestyle and ecology (Carey et al., 2012). Within mammals, the most studied taxon, Seymour and colleagues more recently found neither a 0.67 nor 0.75 value fits the whole of Mammalia (White et al. 2009). They also found that values differ between lineages and these also generally don’t match the 0.67 or 0.75 values, and concluded “no single value...adequately characterizes the allometric relationship between body mass and BMR” (White et al 2009 p.2658). As a consequence of such studies, recent

workers have shifted focus away from comparing support for absolute scaling exponents to isolating the boundaries of variation between exponents (i.e. the “metabolic-level boundaries hypothesis”), accepting that exponents vary between 0.66 and 1 across Animalia depending on taxonomy and physiology (White et al. 2007; Glazier, 2010; 2015). On the basis of these studies, we think it appropriate to not standardise our mass-specific standard metabolic rates.

References

Martins, Emilia P.; Hansen, Thomas F. (1997).

"Phylogenies and the Comparative Method: A General Approach to Incorporating Phylogenetic Information into the Analysis of Interspecific Data".

The American Naturalist. 149 (4): 646–667.

Fuentes-G JA, Housworth EA, Weber A, Martins EP. (2016)

« Phylogenetic ANCOVA: Estimating Changes in Evolutionary Rates as Well as Relationships between Traits. »

The American Naturalist 188 (6) : 615-627.

Seymour R.S., Bennett-Stamper C.L., Johnston S.D., Carrier D.R., Grigg G.C. (2004).

Evidence for endothermic ancestors of crocodiles at the stem of archosaur evolution. *Physiol. Biochem. Zool.* 77:1051–1067.

Summary of response to reviewer 1:

We thank reviewer 1 for their in-depth analysis and consideration of our work. We agree with the majority of their comments and have revised the analyses performed and results presented in our manuscript accordingly. We believe that the revised article is significantly stronger following this process.

References:

- Carey, N., Sigwart, J. D., & Richards, J. G. Economies of scaling: more evidence that allometry of metabolism is linked to activity, metabolic rate and habitat. *J. Exp. Mar. Biol. Ecol.* **439**, 7-14 (2013).
- Glazier, D. S. A unifying explanation for diverse metabolic scaling in animals and plants. *Biol. Rev.* **85**, 111-138 (2010).
- Glazier, D. S. Is metabolic rate a universal ‘pacemaker’ for biological processes? *Biol. Rev.* **90**, 377-407 (2015).
- Symonds, M. R. E., & Elgar, M. A. Phylogeny affects estimation of metabolic scaling in mammals. *Evolution.* **56**, 2330-2333 (2002).
- White, C. R., Cassey, P., & Blackburn, T. M. (2007). Allometric exponents do not support a universal metabolic allometry. *Ecology*, *88*(2), 315-323.
- White, C. R., Blackburn, T. M., & Seymour, R. S. (2009). Phylogenetically informed analysis of the allometry of mammalian basal metabolic rate supports neither geometric nor quarter-power scaling. *Evolution.* **63**, 2658-2667.

Reviewer #2 (Remarks to the Author):

Review of Newham et al “Reptile-like physiology in Early Jurassic stem-mammals” for Nature Communications

Zhe-Xi Luo
 The University of Chicago
 Email: zxluo@uchicago.edu

Appraisal

The paper by Newham and colleagues is highlighting an important question in evolution of mammals. It represents a major progress in the understanding of the life-history characteristics for early mammals.

We are grateful for this further positive endorsement of the importance of our paper.

Extant mammals differ from extant nonmammalian vertebrates in their life-history characters, especially in growth pattern and metabolic rates. This study is aimed to estimate the lifespan - an important life history trait - from the tooth root cementum of the early mammaliaforms *Morganucodon* and *Kuehneotherium*. Early mammaliaforms are already known to be modern mammal-like in many (but not all) of their characters. As lifespans are correlated with growth model and with metabolism rates in modern mammals, and if lifespan can be established for the early mammaliaforms by inference from the cementum increments of fossil teeth, it can help to reveal further insight about the life-history characteristics and the physiological function of these early mammals.

This is an excellent summary of the key points.

To the extent that lifespans can be established for *Morganucodon* and *Kuehneotherium* by first hand observation and with statistical robustness, this study is a giant step forward for characterizing the yearly age and for estimates of metabolic rates for mammaliaforms, which are intermediate between extant mammals, and the pre-mammaliaform cynodonts.

Thanks again for the positive statement.

This paper has a strong research merit - Newham and colleagues have accomplished a sophisticated study by high resolution of synchrotron scans to examine the tooth root cementum increments ("growth layers") of two of the earliest-known mammaliaforms. They have also sampled some impressively large numbers of specimens of *Morganucodon* (36 specimens with discernible growth layers out of 87 specimens examined) and of *Kuehneotherium* (27 specimens showing growth layers out of 119 specimens examined). Following the best practice, the cementum increments are independently counted by three different observers, with robust statistics on the confidence level of the first-hand observation.

We are grateful for the appreciation of the large size of our sample and the statistical approaches we used.

The authors provided some important and very valuable benchmark data:

Morganucodon:

- The longest living specimen or "Longevity champion" is aged to 14 years;
- the mean age at 4.78 years of the entire cohort (the mean of 36 specimens reported in Extended-Data Table 1)

"maximum lifespan" can be calculated at 10 years for *Morganucodon*

Note: The standardized definition of maximum lifespan for demographics studies and biology of senescence is the mean lifespan of the most long-lived 10% of a given cohort. Among the 36

Morganucodon specimens, the 4 specimens of the highest cementum increments are 14, 10, 8 and 8 increments (“years”).

Kuehneotherium:

- The longest living specimen or the “Longevity champion” is aged at 9 years
- the mean age of 4.98 years (calculated as mean of 36 specimens reported in Extended-Data Table 1)

“maximum lifespan” can be calculated at 8.7 years for Kuehneotherium

Note again: the standard definition for maximum lifespan for demographics and biology of longevity is “the mean lifespan of the most long-lived 10% of a given cohort.” Based on the Kuehneotherium specimens with the three highest cementum increments of 9, 8 and 9, out of the 27 Kuehneotherium specimens (Extended Data Table 1).

Synchrotron scans are expensive and the visualizing the data from these scans are extremely time-consuming. The resulted datasets for this study are an enormous accomplishment by itself. I like to compliment Newham and colleagues for their technical skills and time-effort in producing these valuable datasets.

I also found it informative and refreshing that Newham and colleagues used the scaled vascular foramina diameters/volumes of femora as proxies to the blood supply. These are proxies to maximum metabolic rate (MMR) that can be applied to Morganucodon. I found this part of the comparative analysis well reasoned, and technically well justified. The key result is also very reasonable: Morganucodon's MMR is now estimated to be intermediate between the MMR's of extant mammals and the MMR's of extant reptiles. Given the intermediate phylogenetic positions of Morganucodon and Kuehneotherium in cynodonts-mammaliaforms-mammals evolution - this makes sense. Moreover, this result has revealed a new physiological understanding that was not previously available.

The scaling analyses of yearly lifespans of both fossil mammaliaforms and the reference extant mammals with the basal metabolic rate (BMR) and the growth rate constant K (Extended Data Figure 4) show that there are some consistent patterns of these metrics - the estimated growth rate and bmr of mammaliaforms are essentially just like those of extant mammals - if based on lifespans (increments).

However, if scaling the lifespan (years) with the body mass (main text Fig. 4), the story would be different - here the two mammaliaforms are quite different from extant mammals, but more similar to extant reptiles.

We firstly apologise for any confusion caused when reviewing and comparing the regressions between: lifespan versus our metabolic metrics (SMR/K) (originally Extended Data Figure 4; now main text Figure 6); lifespan versus body mass (originally Figure 3b; now Figure 5b); and body mass versus our metabolic metrics (SMR/K) (originally Figure 3 c-d; now Figure 5 c-d). The regression and comparison of the relationship between body mass and lifespan of mammals versus reptiles (originally Figure 3b; now Figure 5b) is used to highlight the disparity between our fossil lifespan estimates and the maximum wild lifespans of extant mammal taxa of comparable body mass. The regressions between maximum lifespan and our two metabolic metrics of SMR and K (originally in Extended Figure 4a/b; now in main text Figure 6a/b) are then used to estimate these metrics from the lifespans of our fossil taxa, following the significant relationships found between lifespan and these metrics in both extant mammals and reptiles. We make these metabolic estimates by placing the mammaliaforms at the point of the regression line corresponding to our estimates of their age and then reading off the resulting SMR or K value. We do this using both the mammal and reptile regression lines in both cases, for robustness of the result. Because of this, in both figure 6a and 6b,

the mammaliaforms appear twice - once each exactly on each of the reptile and mammal regression lines, as required to estimate their SMR and K values with both reptile and mammal derived regression data. We finally plot these estimates of SMR/K against estimated body mass for our fossil taxa in order to compare them to extant taxa of equivalent body mass (originally in Figure 3 c-d; now in Figure 5c-d). For both mammaliaforms, their place on the regression lines (in originally Extended Data Figure 4; now main text Figure 6a/6b) is amongst taxa that are significantly larger than they are, so when corrected for body size (originally in Figure 3 c-d; now in Figure 5c-d), they fall outside of mammalian but within reptilian ranges.

In order to overcome this confusion we have updated the “Long lifespans, low BMR and growth rates” section to now include the regressions between lifespan and msSMR/K as a main figure (originally Extended Data Figure 4a/b; now main text Figure 6a/6b) and explicitly refer to them when discussing the estimated values for our fossil taxa resulting from these regressions. We also added a short passage to the main text to explain this process in brief in addition to it being covered in the methods.

A Major Revision

I think that Newham and colleagues incorrectly interpreted the meaning (definition) of maximum lifespan vs. wildlife lifespan. I offer three inter-related comments and suggestions for revision:

Suggestion 1- Newham and colleagues defined the maximum lifespan here on a singular “long-life champion” specimen with 14 increments for Morganucodon, out of the 36 specimens of this Morganucodon sample. However, the maximum lifespan is defined differently in demographic and senescence studies - In demographics, and in senescence and longevity-related biological studies, the “maximum lifespan” is defined as the mean lifespan of the most long-lived 10% of a given cohort, not a “longevity champion age” (or “Guinness World Record”) -14 increments in one specimen - as the authors picked for Morganucodon. Given the well documented data across an extensive sample of Morganucodon teeth, the “maximum lifespan” should be recalculated as the mean increment of the top 4 specimens of a sample of 36 specimens.

To pick a “longevity champion” at 14 increments is far less justifiable statistically, as it deviates from the standard of the maximum lifespan in demographic studies of extant mammals (defined as the “longest-10%-mean.”) I really urge that the authors follow the “mean of top 10% of longest living individuals of the given cohort”, and use the top 4 specimens for the mean of this metric. If so, this change would lead to a revision of 14 increments (longevity champion) to 10 increments (the mean of four specimens with most increments) for Morganucodon. Similarly the 9 increments of Kuehneotherium would be revised to 8.7 increments.

This would be the first step. Then authors should place 10 increments for Morganucodon and 8.7 increments for Kuehneotherium in regression analyses (Figure 4B).

Unfortunately, we cannot follow this suggestion for biological reasons, even if our fossil samples represented cohort populations, which they do not. **Although it is occasionally stated on several websites that the 10% longest living cohort is the most widely used metric, this is only rarely applied in published datasets, especially large-scale evolutionary comparative ones, mainly because of the small available sample sizes (i.e. population level longevity studies of different species) for this metric. The most common datapoint available is the known longest living single individual. For example, in the online AnAge database, of the 865 mammalian taxa with longevity estimates deemed by the database as at least “acceptable”, the maximum captive lifespan of over half of the sample is specifically represented by oldest known single individuals, or “longevity champions”. Further, studies that we have reviewed (Andziak et al., 2005; Willcox et al., 2007), that use the 10% longest living cohort metric, base this value upon longitudinal studies of single populations that monitor at least one generation through their entire lifespan. This is not possible for our fossil**

sample as they do not represent single cohorts in the way required. Our sample is comprised of fossils from one (*Morganucodon watsoni*) or several (*Kuehneotherium*) different fissure fill deposits. It is unclear how individual specimens are related both geographically and temporally and so this sample does not fulfil the requirements to use the 10% longest living cohort metric. In contrast to cohort-based metrics such as 10% longest living cohort and average/median life expectancy, that may change dependent on living conditions and intra/inter-population level genetic variation, maximum lifespan is a stable characteristic of a species (Dong et al., 2016; Bozek et al., 2017). This value is a by-product of genetic programs for life history such as development, growth and reproduction, that determine species-specific lifespan (Dong et al. 2016). These are the physiological properties of interest to our research. Therefore, we believe retaining our maximum lifespan estimates from our longevity champions is justified, as this both provides the most fitting comparison for data representing the overwhelming majority of extant taxa, and is the most suitable metric for our sample, given its provenance.

Suggestion 2 - After the revision of the maximum lifespan as “the mean of longest-living 10% of the total cohort of the cementum sampling, if Newham and colleagues still prefer to use “maximum lifespan” as a key metric for comparative discussion, then they should also revise and recompose the comparative data of extant mammals. They should use the captivity longevity of extant mammals against body mass of extant mammals in regression (Figure 4B). The reason for this suggestion is that the captivity longevity is a better representation of the realized lifespan, while the maximum lifespan cannot be reliably observed in real wildlife population. The AnAge database (<https://genomics.senescence.info/species/>) is a live and curated database, and the most widely used in demographics and senescence studies. This mainstream database shows the maximum lifespan in captivity as the default maximum age, and the only formally listed yearly age.

Newham and colleagues did not use this default age estimate from AnAge database for the majority of mammals in their reference dataset. Instead, for the wildlife lifespan estimates of the comparative mammal data, they optioned to pick much younger age estimates from the wildlife spans from other studies.

First - If “maximum lifespan” (best-recorded increments of a specimen) is used for *Morganucodon* and *Kuehneotherium*, then for the reference data of extant mammals, the maximum lifespans in captivity should also be used. This is to be biologically consistent.

Second - In demographics of extant mammals, the “longest-documented lifespan in captivity” is the most reliable data, and the most widely available of the various “lifespan metrics.” For the AnAge database (genomics.senescence.info/species/), the longest life span in captivity is the default metric. Further, the AnAge database provides careful comments on the variable nature of “wildlife” lifespan. The data on variation of lifespan (other than known max lifespan by captivity) are discussed in the footnotes, and discussion on reliability ranking is sourced to the literature. The Max Planck database on longevity records have lists of both wildlife lifespan and the captivity longevity, but its taxonomic coverage on longevity in captivity is more extensive than “wildlife lifespan” for the terrestrial (non-cetacean and non-chiropteran) mammals. In sum, the captivity lifespan is the hard data, and more available, and less susceptible to uncertainty.

The AnAge website provides an extremely useful repository for longevity and body mass data of extant animals, and we have incorporated data on captive lifespan from it into our new dataset compiled to compare wild and captive lifespans of extant mammals and reptiles. When we compare this captive maximum lifespan data from the AnAge database, Max Planck and Scharf et al. (2015) databases with wild lifespan data for the same taxa, we see average increases in captive lifespan compared to wild lifespan (SI Figure 4a) for both mammals and reptiles. These increases are significant when tested with ANCOVA (mammal average increase from wild to captive lifespan of

4.67 $F = 71.54$, $p < 0.001$; reptile average increase from wild to captive lifespan of 6.41 $F = 21.66$, $p < 0.001$). As shown in SI Figure 4a, differences are most extreme for small bodied taxa of comparable body mass to those predicted for our fossils, as previously reported in the literature (Tidiere et al. 2016). These significant differences suggest that wild lifespan cannot be fairly compared with captive lifespan. However, our fossils can only provide an estimate of wild lifespan, which was therefore used in the manuscript in order to avoid bias. The referee is however correct that the captive data is generally more robust. In order to compare our 'wild' lifespans estimated for our fossil taxa with the maximum captive lifespans of extant mammals and reptiles, we need to in-turn provide an estimate for their 'captive' lifespans.

To do so, we used the average increase in captive lifespan calculated above for both mammals and reptiles to estimate 'captive' lifespan for *Morganucodon* (18.67 and 20.41 respectively) and *Kuehneotherium* (14.67 and 16.41 respectively) (SI Figure 4). We then estimated their SMR/K using the relationship between maximum captive lifespan and these metrics in the same manner, and with the same extant mammals and reptiles as used for estimating SMR/K from wild lifespans (SI Figure 5). This is mentioned in the main text and the data, results and discussion of this is included in the SI (Supplementary discussion - "Comparisons between fossil lifespan estimates and captive lifespans of extant taxa"). Confirming the results of our 'wild' maximum lifespan analyses, *Morganucodon* and *Kuehneotherium* have 'captive' maximum lifespans within the range of extant reptiles and considerably higher than similarly sized extant mammals (SI figs 4,5), and the 'captive' maximum lifespan of our fossil taxa provide significantly lower SMR/K estimates than extant mammals of comparable size, within the range of extant reptiles (SI figure 5). Overall, these new analyses should make the results of the study more robust, but also highlight areas that will need more attention in the future.

Third - In many histological studies of cementum growth layers that have traced many individuals across a range of ages for a species, the specimens are mostly (if not entirely) from captive animals, or from animals that were wild-caught, but kept captive. It appears from the tables and graphics of the historical studies by Klevezal and her colleagues that these cementum increment studies are not based on real wildlife populations.

The direct data for the Newham et al. study are the cementum increments. As some of the best available series of cementum increments in extant mammals are sampled from kept populations, it is better to compare the fossil increment data with the extant mammal cementum increments data from captive/kept populations. This is to be consistent in nature of cementum data of both extant mammals, and the fossil species involved.

Although several studies of captive mammal cementum have been published (see Klevezal, 1995 for a review of these studies), these authors have generally concluded that the incrementation of cementum in captive mammals is of lower quality than wild mammals. As cementum is known to be directly affected by seasonal changes in dietary quality (Leiberman, 1994), this has been suggested to be due to a dampening of the effects of seasonal environmental fluctuations and consistent food supply in captive populations. Greater accuracy has been provided by monitored free-roaming populations of cervids such as reindeer, who are regularly monitored, but experience the full magnitude of climatic change through the year (Pasda, 2006). It is now estimated that over 2.5 million individual wild animals have been aged by professional laboratories (e.g. <https://matsonslab.com/>), primarily for conservation purposes. Finally, as cementochronology is primarily used as a lifespan estimate for individuals of unknown life span, it is rarely used and published for captive populations compared to wild populations.

Suggestion 2 - An alternative way to revise - a better and easier way in my opinion - would be to use the statistical mean of the cementum increments of the 36 Morganucodon specimens, which is 4.78 increments. The authors can then plot this number into Figure 4B for the regression of the wildlife lifespan of extant mammals against body mass of extant mammals.

This way, the authors can be biologically consistent parsing their data for comparison, yet they do not need to go through the tedious and more cumbersome revision of all “wildlife lifespan” data of extant mammals that have already collected and programmed into the current version of the paper. I believe that this alternative is much better for several considerations:

First - the statistical mean (4.78 increments for Morganucodon, 4.98 increments for Kuehneotherium) is more reliable for estimating yearly age, than “maximum” increment (14 for Morganucodon and 9 for Kuehneotherium). Several cementum age estimate papers on large sample sizes have consistently shown that annual c-increments are correlated with yearly age only in young individuals. But the increments can deviate significantly from yearly age in older adults of the same species. Grau et al (1970) study on raccoons (*Procyon lotor*) (ref 73) demonstrated this decline of accuracy of annuli-year correlation in older specimens. Grau (1970) shows that the method works well before 4 years of age, but cementum age estimate is rapidly declining in accuracy for individuals older than 4 years of age. For 5 years or older, cementum rings are only 70%-50% accurate for estimating yearly age. Kay and Cant (1988) (ref 76) on *Maraca mulatta* in a very thorough study - it has demonstrated the correlation of cementum “annuli” with yearly age can vary significantly over the lifespan of individual specimens. In older specimens there is an increasingly wider scattering of datum points on either side of the regression line (Kay and Cant 1988: figure 3) - wider variation (= less accuracy) in older adults that are 12 years or beyond. Further, the older individuals have a tendency to show different annulus counts on the left versus right first molars of the same individual (Kay and Cant 1988: table V).

To put it more simply - when individuals of mammals approach their “maximum lifespan,” it is progressively less reliable to count on the cementum “annuli” as a proxy to yearly increments.

For this reason, it is statistically sound and biologically more justifiable if one uses the mean of cementum increments across a sample, such as the 36 specimens counted for their increments for Morganucodon, and 27 specimens for Kuehneotherium in Extended Data Table 1.

Second - the statistical mean of increments of the fossil teeth can be regarded as the peak mortality age of the populations of the two mammaliaforms. This is to take the mean of increment counts as a proxy to the peak age of death. This proxy is relevant to demographics of these fossil taxa in real paleoecological context. Given the large sample size and the fact that teeth of both Morganucodon and Kuehneotherium were collected from multiple quarries/localities, this is a biologically justifiable approach under the time-average principle of paleoecology.

Third - it makes the best sense to compare the peak mortality (mean of the increments across the entire sample) of Morganucodon/Kuehneotherium to the wildlife lifespan data of extant mammals in the reference dataset. The means of these samples are more realistic in terms of paleoecology, and more comparable to extant wildlife mammal lifespan in the context of their ecology. This is better than picking a singular specimen as the symbolic “longevity champion.”

Fourth - A great strength of this work is that it has amassed a significant body of data (36 increment counts of Morganucodon and 27 counts of Kuehneotherium). It would be more powerful to fully capitalize these datasets and to use a metric with robust statistics.

I hasten to add - this is a much easier way to revise this manuscript because the statistical means of cementum increments are already available: 4.78 annuli for Morganucodon and 4.98 annuli for Kuehneotherium. There is no need to vet and to revise the entries of raw data from extant mammals, and re-compose the dataset for the regression analysis.

My expectation is that the mean of 4.78 increments Morganucodon and 4.98 for Kuehneotherium are closer to the lifespans of extant mammals, than the previous estimates of 14 for Morganucodon and 9 for Kuehneotherium. Thus the estimates on basal metabolic rate (BMR, or RMR) for these two mammaliaforms would be more likely intermediate between the extant reptiles and extant mammals. This would be more consistent with the estimate on maximum (aerobic) metabolic rate (MMR) from the femoral vascular channels as proxy to blood vessel volumes.

As previously stated, it is not statistically sound to treat our fossil samples as a single cohort of each taxon. Further, in addition to the taphonomic and time-averaging biases affecting our sample, we have added our own selection bias by only selecting lower m2 teeth with obviously preserved cementum. Maximum known lifespan is resistant to these biases, compared to other lifespan metrics, as it is primarily reflected by sample size rather than sample origin (i.e. maximum lifespan values can only increase with sample size). While certainly affected by the maximum known lifespan of a sampled population, mean/median lifespan is not a direct representative of the physiological maximum that an organism can live to. Mean/median lifespan values are likely to be affected by extrinsic factors on a single or range of populations including disease, predation, dietary and climate effects which may bias their values when analysing the relationship between lifespan and metabolic metrics.

Although the cited early validation studies suggest a reduction in accuracy of cementochronology for age estimation with increased lifespan, more recent studies utilising an increasing array of visualisation techniques and computer vision approaches have achieved greater accuracies (Wittwer-Backofen et al., 2004; see Naji et al., 2016 for a review of the most-up-to-date procedures and advancements). Professional laboratories now use cementochronology to age hundreds of thousands of individual animals per year, over 2.5 million individuals cumulatively (see www.matsonlab.com) and age wild samples of a wide range of taxa with maximum ages of up to 42 years and accuracies of \pm one year. Recent validation studies of cementochronological ageing of human teeth have also produced high accuracies for individuals aged over 90 years old (Condon et al., 1986; Wittwer-Backofen et al., 2004). Although most validation studies do show a varying loss in accuracy with increasing age, this is typically represented by an underestimation of known age, which would act to bias the estimated maximum lifespans of our mammaliaforms downwards from their true age, in comparison to the wild lifespans in our comparative dataset recorded with other methods. We also believe that due to our sophisticated imaging techniques and high inter-observer precision our use of maximum lifespan estimates is not undermined by a reduction in accuracy with age.

A Clarification

Can the authors clarify a discrepancy that I have noticed between Text Figure 4 and Extended Data Figure S4?

Extended Data Figure S4 shows consistent patterns of Morganucodon/Kuehneotherium and the extant mammals in the standard metabolic rate plus the growth rate constant K of extant mammals, with respect to their cementum-based estimates (fossils) or actual yearly ages (extant). These regression analyses seem to suggest that the growth rate and smr of the two mammaliaforms are in the similar range as those of extant mammals, if scaled according to their ages (=cementum increments for fossils). This (ED Fig. 4) result is sensible.

However, the results are quite different in scaling of the standard metabolic rate and growth rate constant K of Morganucodon/Kuehneotherium and extant mammals with respect to their body masses, as illustrated in text Fig. 4. Here the two mammaliaforms are different from extant mammals, but are more similar to extant reptiles. The main narrative of the paper is based on Fig. 4, but not on Extended Data Fig 4.

Why are placements of Morganucodon and Kuehneotherium are so different in these two sets of regression analyses? I do not disagree with these, but I am puzzled. I suggest that authors provide a discussion to clarify this in Supplementary Information.

As previously stated, we take full responsibility for this confusion and apologise. As noted above, the placement of our fossil taxa along the regression line between lifespan and SMR, and lifespan and K for both extant reptiles and extant mammals (originally in Extended Data Figure 4a/b; now in main text Figure 6a/b) is because these regressions are used to estimate the values of SMR and K from the lifespans of our fossil mammals. These estimates then differ from the placement of comparably sized extant mammal taxa when they are regressed against body mass, highlighting the significant difference between values predicted for our fossil taxa, and those known for comparably sized mammals.

Correction on Extended Data Figure 2.

The dicynodonts are a clade. But the four dicynodonts mentioned in this figure - Kawingasaurus, Oudenodon, Lystrosaurus, Moghreberia - are tagged onto the backbone of the phylogenetic tree in four different nodes. This could be misconstrued that the authors would interpret that dicynodonts are paraphyletic (the authors clearly did not intend this) and that these dicynodonts have different phylogenetic positions, with regards to Thrinaxodon. I understand well that this figure is designed to summarize all synapsid taxa that have been inferred for physiological functions, as mapped on geological time scale. But if you use phylogeny at all, you must somehow combine all four dicynodonts into the dicynodont clade, and dicynodonts as a whole should have a single node on the synapsid phylogeny.

Extended Data Figure 2. Typo - For Morganucodon - the “determinant” growth should be “determinate” growth.

Figure 4 - typo - Figure 4d, Y-axis label - Growth rate constant K (days⁻¹). “constant” should be “constant”

Reference 68. The last names of all authors got lost in this listed reference. The name of the journal is also wrong. I think this is Felisa A. Smith, Kathleen S. Lyon, et al. (2003) Ecology.

We thank Luo for his thorough review of all our figures, captions and references and we have adjusted these accordingly.

Summary

I strongly support that Nature Communications should publish this important paper. I just like to suggest that Newham and colleagues reconsider how they would best parse the comparative data, so that the comparison is biologically justifiable. They should use the mean of increments of the entire sample of mammaliaforms, in comparison with the wildlife lifespan data of extant mammals, as these are ecologically comparable. Or they can continue to use the mean of maximum cementum increments (14 for Morganucodon, 9 for Kuehneotherium), but to do so they would need to re-vamp the comparative extant mammal dataset for the maximum lifespans in captivity

Summary of response to reviewer 2:

We thank Luo for his in-depth and well considered review and praise for our work. We value his suggestions for further work to refine our data and comparisons with extant taxa. Following consideration of Luo's suggestions, we now mention wild versus captive data in the main text and provide analysis and discussion in the supplement of our revised article, including an estimate of 'captive' lifespan for our fossil taxa based on the significant difference found between wild and captive lifespan for extant taxa. As with our 'wild' estimates of maximum lifespan, captive estimates provide significantly lower values of SMR and K than extant mammals, supporting our initial conclusions. We believe that this approach is the most statistically robust way of accommodating for captive data in our analysis; both acknowledging its importance for robustly comparing our estimated ages and overcoming the limitations of the lack of suitable data for our fossils.

We have additionally made all requested formatting changes and agree that our article is now substantially improved thanks to your peer review.

References.

- o Andziak, B., O'Connor, T. P., & Buffenstein, R. Antioxidants do not explain the disparate longevity between mice and the longest-living rodent, the naked mole-rat. *Mech. Ageing Dev.* **126**, 1206-1212 (2005).
- o Bozek, K., Khrameeva, E. E., Reznick, J. et al. Lipidome determinants of maximal lifespan in mammals. *Sci, Rep.* **7**, 5-14 (2017).
- o Dong, X., Milholland, B., & Viig, J. Evidence for a limit to human lifespan. *Nature.* **538**, 257-259 (2016).
- o Naji, S., Colard, T., Blondiaux, J., Bertrand, B., d'Incau, E. et al. Cementochronology, to cut or not to cut? *Int. J. Paleopathol.* **15**, 113-119 (2016).
- o Pasda, K. Assessment of age and season of death of West Greenland reindeer by counting cementum increments in molars. *Documenta Archaeobiologiae*, **4**, 125-140 (2006).
- o Tidière, M., Gaillard, J. M., Berger, V., Müller, D. W., Lackey, L. B., et al. (2016). Comparative analyses of longevity and senescence reveal variable survival benefits of living in zoos across mammals. *Scientific reports*, **6**, 36361.
- o Willcox, B. J., Willcox, D. C., Todoriki, H., Fujiyoshi, A., Yano, K., He, Q., Curb, J. D., & Suzuki, M. The diet of the world's longest lived people and its potential impact on morbidity and life span. *Ann. NY Acad. Sci.* **1114**, 434-455 (2007).

Reviewers' Comments:

Reviewer #1:

None

Reviewer #2:

Remarks to the Author:

Reptile-like physiology in Early Jurassic stem-mammals
(first manuscript Nov 2019; revised manuscript June 2020)
Elis Newham et al. (Correspondence author: Ian Corfe)

Reviewer: Zhe-Xi Luo/UChicago (zxluo@uchicago.edu)

This paper has two strong points: truly excellent data on fossils, a set of elaborated analyses to tie the age estimates (cementochronology) to basal metabolic rates (both Morganucodon and Kuehneotherium), and to tie the size of vascular foramina of femurs to the maximal metabolic rate (for Morganucodon).

I fully endorse the key conclusion that Morganucodon is intermediate between extant mammals and extant reptiles in the maximum metabolic rate inferred from femoral vascular foramina. I am willing to listen to the perspective from the authors that Morganucodon and Kuehneotherium were more like extant reptile, than extant small mammals in basal metabolic rate.

The paramount importance at this juncture in the studies of early evolution of mammals is for this to be published as a great case-work for an approach (cementochronology for age estimates) for early mammals, and its significant body of data for estimating the likely ages of Morganucodon (on 36 specimens, out of 87 specimens examined) and Kuehneotherium (27 specimens, out of 119 specimens examined). I much admired this accomplishment.

In response to my review comments, Newham and colleagues revised several parts of the manuscript. Figures are also re-arranged in the revision, and the new text Figure 2 is good in contextualizing some complex prior approaches and ideas in estimating metabolic physiology of pre-mammalian synapsids. This is setting up a nuanced tone in presenting the questions, and analyses of the current study.

The authors clearly disagreed with my comment on how to re-consider the complex comparative data of the ages of extant mammals. They offered a clarification that the maximum lifespan (a crucial criterion) is herein defined as the longevity champion of a sample, but not the average of the top 10% of this sample. I think this issue is a nuanced problem that could be a step-stone for future studies. And I can accept the authors' preference in treating this, in the current manuscript.

I support for Nature Communications to publish this paper in current form. I like to congratulate Newham and colleagues for such an exquisite work, and thoughtful work.

REVIEWERS' COMMENTS:

Reviewer #2 (Remarks to the Author):

Reptile-like physiology in Early Jurassic stem-mammals
(first manuscript Nov 2019; revised manuscript June 2020)
Elis Newham et al. (Correspondence author: Ian Corfe)

Reviewer: Zhe-Xi Luo/UChicago (zxluo@uchicago.edu)

This paper has two strong points: truly excellent data on fossils, a set of elaborated analyses to tie the age estimates (cementochronology) to basal metabolic rates (both Morganucodon and Kuehneotherium), and to tie the size of vascular foramina of femurs to the maximal metabolic rate (for Morganucodon).

I fully endorse the key conclusion that Morganucodon is intermediate between extant mammals and extant reptiles in the maximum metabolic rate inferred from femoral vascular foramina. I am willing to listen to the perspective from the authors that Morganucodon and Kuehneotherium were more like extant reptile, than extant small mammals in basal metabolic rate.

The paramount importance at this juncture in the studies of early evolution of mammals is for this to be published as a great case-work for an approach (cementochronology for age estimates) for early mammals, and its significant body of data for estimating the likely ages of Morganucodon (on 36 specimens, out of 87 specimens examined) and Kuehneotherium (27 specimens, out of 119 specimens examined). I much admired this accomplishment.

In response to my review comments, Newham and colleagues revised several parts of the manuscript. Figures are also re-arranged in the revision, and the new text Figure 2 is good in contextualizing some complex prior approaches and ideas in estimating metabolic physiology of pre-mammalian synapsids. This is setting up a nuanced tone in presenting the questions, and analyses of the current study.

The authors clearly disagreed with my comment on how to re-consider the complex comparative data of the ages of extant mammals. They offered a clarification that the maximum lifespan (a crucial criterion) is herein defined as the longevity champion of a sample, but not the average of the top 10% of this sample. I think this issue is a nuanced problem that could be a step-stone for future studies. And I can accept the authors' preference in treating this, in the current manuscript.

I support for Nature Communications to publish this paper in current form. I like to congratulations Newham and colleagues for such an exquisite work, and thoughtful work.